# Concepts' Information Bottleneck Models

## Abstract

Concept Bottleneck Models (CBMs) promise interpretable prediction by forcing all information to flow through a human-understandable "concept" layer, but this interpretability often comes at the cost of reduced accuracy and concept leakage. To solve this, we introduce an explicit Information Bottleneck regularizer on the concept layer—penalizing $I(X; C)$—to encourage minimal yet task-relevant concept representations. We derive two variants of this penalty and integrate them into the standard CBM training objective. Across six model families (hard/soft CBMs trained jointly or independently, ProbCBM, AR-CBM, and CEM) and three benchmark datasets (CUB, AwA2, aPY), IB-regularized models consistently outperform their vanilla counterparts—narrowing and in some cases closing the accuracy gap to unconstrained black-box networks. We further quantify concept leakage with two metrics (Oracle Impurity and Niche Impurity Scores) and show that IB constraints reduce leakage significantly, yielding more disentangled concepts. To assess how well concept sets support test-time corrections, we employed two intervention metrics (area under the intervention-accuracy curve and average marginal gain per intervened concept) demonstrating that IB-regularized CBMs retain higher intervention gains even when large fractions of concepts are corrupted. Our results reveal that enforcing a minimal-sufficient concept bottleneck improves both predictive performance and the reliability of concept-level interventions, thereby closing the accuracy gap of CBMs while improving their interpretability and ability to intervene.

## 1  Introduction

In many real-world settings, from medical diagnosis to autonomous driving, models must do both: make accurate predictions and provide explanations that humans can trust. Consequently, explainable AI seeks to peel back the curtain on opaque machine learning systems, boosting trust, accountability, and safety by exposing hidden biases and errors. We categorize explainable models into four groups. *Post-hoc techniques* explain black-box models after training, using interpretable approximations or feature attributions [19]. *Model-agnostic methods* treat the model as a black box, analyzing inputs and outputs. These include *local interpretability methods*, which explain individual predictions [16, 17], and *global interpretability methods*, which provide broader insights [2, 7]. Finally, *self-explainable models* are inherently interpretable, requiring no additional techniques. This work champions *self-explainable models* because they deliver structured, inherent explanations and seamless debugging, making them a promising alternative to other approaches.

Concept bottleneck models (CBMs) [12] are a self-explainable approach that modifies neural network training by introducing intermediate, human-understandable concept labels, enabling predictions to be based on these concepts. CBMs aim to explain final decisions through these interpretable concepts and allow users to correct concept predictions to refine the model's outputs. Their advantages

include higher robustness to covariate shifts and spurious correlations when predictions rely solely on concepts. However, CBMs often underperform compared to black-box models. Additionally, they suffer from *concept leakage* [13, 14], where irrelevant information is encoded in concept activations, affecting both interpretability and the effectiveness of test-time interventions.

Rather than redesigning CBM architectures or enriching concept embeddings as done in previous works [8, 10], we take a simpler, information-theoretic approach: we impose an Information Bottleneck [1, 20] penalty directly on the concept layer. By penalizing the mutual information between inputs and concepts while still maximizing their informativeness about the target, our method suppresses spurious signals, closes the accuracy gap to black-box models, and yields more reliable, intervenable concepts. We demonstrate that adding this bottleneck to diverse CBM variants consistently boosts performance and reduces concept leakage.

The main contributions of this work are two-fold: (i) a new CBM loss (regularization) that exploits the Information Bottleneck (IB) framework providing a significant improvement compared to both vanilla and advanced CBMs, and (ii) a demonstration that CBMs that are IB-regularized achieve better predictions, show less concept leakage, and are more robust to interventions than their non-regularized counterparts.

## 2 Related work

### 2.1 Concept Bottleneck Models

**CBMs.** A CBM [12] is defined as $\hat{y} = f(g(x))$, where $x \in \mathbb{R}^D$, $g \colon \mathbb{R}^D \to \mathbb{R}^k$ is a mapping from raw feature space into the lower-dimensional concepts space, and $f \colon \mathbb{R}^K \to \mathbb{R}$ is a mapping from the concepts to the target variable. For training this model composition, a dataset of triplets $\{(x_i, c_i, y_i)\}_{i=1}^N$ is needed, where $c_{(\cdot)}$ stands for the ground-truth concepts labels which should be produced by $g$. The CBM could be trained independently, sequentially, or jointly [12]. Intuitively, when training a CBM, one is introducing human-understandable sub-labels (concepts) which are more primitive and general than the target, and then builds a model predicting the target based solely on those explainable concepts. However, despite these benefits, CBMs often lag behind unconstrained "black-box" models in prediction performance.

To bridge this gap, Concept Embedding Models (CEM) [3] learn two vectors for each concept ("active" and "inactive"). Such approach has increased target accuracy, but requires additional regularization algorithm called 'RandInt' for CEM to be able to effectively utilize test-time interventions. Moreover, the analysis of information flow done in CEMs suggests that information between inputs and concepts is monotonically increasing without any compression.

Our proposal, unlike CEM, maintains the original model concept representation space and regularizes it through our concept information bottleneck regularization. Since we incorporate mutual information constraint into loss function, we can apply our regularization to different models (as demonstrated in our experiments).

**Probabilistic CBMs.** Probabilistic approaches have been explored recently as well to better model the concepts, e.g., ProbCBMs [10] or ECBMs [23], which predict distribution of concepts and use anchor points for class mapping. Similarly by introducing inductive biases, previous work [15, 24] can extract the concepts without annotations. In this work, we do not utilize these anchor points, since they increase inference costs and introduce a new hyper-parameter to tune at fitting stage. We do use a variational approximation over our proposed concepts' information bottleneck to predict concepts.

**Post-hoc CBMs.** Another line of work investigates the transformation of any pre-trained model into a CBM. Side-channel CBMs [8] allow the information to flow through a side concept bottleneck. Recurrent CBMs [8] predicts concepts one after the other using information about previous concept predictions. However, side-channel CBMs have lower intervenability, and recurrent ones break the disentanglement of concepts. Post-Hoc Concept Bottleneck Models (PCBM) [25] use image embeddings from a pre-trained CNN's penultimate layer activations. However, these models perform well only after residual connections are added, moreover, concepts classifiers are learned post-hoc on top of frozen embeddings, which makes it impossible to alter the pre-concept representations learning target. This residual information flow may damage both interpretability and intervenability.

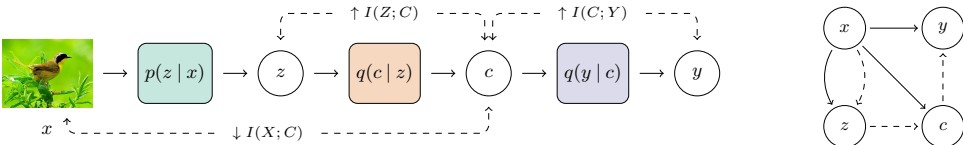

**Figure 1:** Our proposed CIBMs pipeline. The image is encoded through $p(z \mid x)$, which in turn encodes the concepts with $q(c \mid z)$, and the labels are predicted through $q(y \mid c)$. These modules are implemented as neural networks. We introduced the IB regularization as mutual information optimizations over the variables as shown in dashed lines.

**Figure 2:** Our generative model $p(y \mid x)p(c \mid x)p(z \mid x)p(x)$ (solid lines), and its variational approximation $q(y \mid c)q(c \mid z)q(z \mid x)q(x)$ (dashed lines).

**Concept Leakage.** One problem with CBMs is the leakage of information into the concepts [13, 14], regardless of being soft (taking values between $[0, 1]$) or hard (values clipped to $\{0, 1\}$). Margeloiu et al. [14] argue that the CBMs desiderata is met for independent training only: for joint and sequential a CBM learns more information about the raw data than just that presented in the concepts. Thus, concepts are not used as intended. Developing the idea of tracking concepts predictions, Margeloiu et al. [14] apply saliency methods to backtrace concepts to input features and find that for neither training method of the three derive concepts from something meaningful in the input space. Similarly, the Oracle and Niche Impurity Scores [4] were proposed to further understand the level of leakage. Conversely, we hypothesize that by compressing the concepts and the data, and, simultaneously, maximally expressing the labels and concepts through their respective variables, we could obtain better concepts and representations. Our experiments, support this hypothesis by the different improvements across a diverse set of tasks.

## 2.2 Information Bottleneck

Tishby et al. [20] introduced the information bottleneck (IB) as the minimization of the functional $\mathcal{L}_{\text{IB}} = I(X; Z) - \beta I(Z; Y)$, where $I(\cdot; \cdot)$ is the mutual information, $\beta$ is the Lagrange multiplier, $X$, $Y$ and $Z$ are random variables that represents the data, labels, and latent representations, respectively. The motivation behind the bottleneck is to "squeeze" the relevant information about target $Y$ from $X$ into a compact representation $Z$ while minimizing the information about input $X$ in $Z$—so that the representations are free of irrelevant information from $X$. The IB's authors have also posited that good generalization is connected with memorization-compression pattern. This is the behavior in which $I(Z; Y)$ increases during the whole training time, while $I(X; Z)$ increases at first (memorization) and then decreases at later iterations (compression).

Alemi et al. [1] extended the IB framework to deep neural networks by doing a variational approximation of latent representation. And, Kawaguchi et al. [9] analyzed the role of IB in estimation of generalization gaps for classification task. Their result implies that by incorporating the Information Bottleneck into learning objective one may get more generalized and robust network. Unlike this previous work that studied the IB for the data and the labels, we introduced another predictive variable, the concepts, and derive an upper bound that links common predictors and the ground truth into a regularizer that enforces the memorization-compression dynamics. Moreover, we show that the concepts' information bottleneck can be used in common CBM approaches through a mutual information estimator as well.

## 3 Concepts' Information Bottleneck

Concept Bottleneck Models (CBMs) aim for high interpretability by introducing human-understandable concepts, $C$, as an intermediary between latent representations, $Z$, and the labels $Y$. To preserve the interpretability at the heart of CBMs, our objective seeks to minimize $I(X; C)$—the mutual information between inputs and concepts. Thereby, it ensures that concepts remain meaningful and free from irrelevant data, while addressing concept leakage by controlling the information flow directly at the concept level, rather than at the more abstract latent space, $Z$. Simultaneously, we aim to maximize the expressivity of the concepts about the labels, $I(C; Y)$, as well as the one of the latent representations and the concepts, $I(Z; C)$. Our initial objective is $\max I(Z; C) + I(C; Y)$, s.t. $I(X; Z) \leq I_C$, where $I_C$ is an information constraint constant, that equivalently is the maximization of the functional of the concepts' information bottleneck (CIB)

$$\mathcal{L}_{\text{CIB}} = I(Z; C) + I(C; Y) - \beta I(X; Z), \tag{1}$$

where $\beta$ is a Lagrangian multiplier. This formulation ensures a strong connection between latents, $Z$, and the concepts, $C$. This means that one wants $Z$ to be maximally useful in shaping the concepts $C$, while also ensuring that the concepts are informative about the target.

Moreover, in the CBMs formulation, the concepts come from processing the latent representations, i.e., $c = h(z)$. Thus, due to the data processing inequality, $I(X;C) \leq I(X;Z)$, we can bound of the concepts' information bottleneck loss (1) as $I(Z;C) + I(C;Y) - \beta I(X;C) \geq I(Z;C) + I(C;Y) - \beta I(X;Z)$.

Our objective is to maximize the upper bound of the concepts' information bottleneck

$$\mathcal{L}_{\text{UB-CIB}} = I(Z;C) + I(C;Y) - \beta I(X;C).^{1} \tag{2}$$

We depict our general framework in Fig. 1. We posit that by compressing the information between the data, $X$, and the concepts, $C$, instead of the latent representations, $Z$, we can control the redundant information of the data within the concepts. Consequently, we can obtain more interpretable concepts instead of first compressing the latents and then obtaining the concepts from them. We hypothesize that this compression also prevents data leakage from the data into the concepts that commonly happens when the concepts are processed through the latents alone. Another interpretation of this process is the compression of the information between the data and the concepts through the marginalized latent representations. Thus, we are obtaining a more robust compression since we compute it through all possible latent representations that lead to that concept.

We propose two implementations of our framework by exploring different ways of solving the mutual information based on a variational approximation of the data distribution. We show our modeling assumptions in Fig. 2.

## 3.1 Bounded CIB

We can consider the upper bound to the concept bottleneck loss (2) in terms of the entropy-based definitions of the mutual information. Then, by using a variational approximation of the data distribution, we bound it by

$$\mathcal{L}_{\text{UB-CIB}} \leq H(Y) + (1-\beta)H(C) + H\left(p(y\,|\,c), q(y\,|\,c)\right) + (1+\beta)\underset{p(z)}{\mathbb{E}}H\left(p(c\,|\,z), q(c\,|\,z)\right), \tag{3}$$

$$\mathcal{L}_{\text{UB-CIB}} \leq (1-\beta)H(C) + \underset{p(c)}{\mathbb{E}}H\left(p(y\,|\,c), q(y\,|\,c)\right) + (1+\beta)\underset{p(z)}{\mathbb{E}}H\left(p(c\,|\,z), q(c\,|\,z)\right). \tag{4}$$

We detail this derivation in Appendix A. We can maximize the concepts' information bottleneck by minimizing the cross entropies of the predictive variables, $y$ and $c$, and their corresponding ground truths and by adjusting the entropy of the concepts—cf. Fig. 2. The simplified upper bound of the concept information bottleneck is

$$\mathcal{L}_{\text{SUB-CIB}} = (1-\beta)H(C) + \underset{p(c)}{\mathbb{E}}H\left(p(y\,|\,c), q(y\,|\,c)\right) + (1+\beta)\underset{p(z)}{\mathbb{E}}H\left(p(c\,|\,z), q(c\,|\,z)\right). \tag{5}$$

We denote the models that were trained using this bounded concept information bottleneck (5) by $\text{IB}_B$. To implement it, we need to estimate the entropy of the concepts distribution $p(c)$. We give details of this estimator in Appendix B.2.

## 3.2 Estimator-based CIB

Another way to obtain a bound over the concept information bottleneck (2) is to only expand the conditional entropies that are not marginalized (A.1) to avoid widening the gap in the bound, i.e.,

$$\mathcal{L}_{\text{UB-CIB}} = H(Y) + H(C) + \underset{p(c)}{\mathbb{E}}H\left(p(y\,|\,c), q(y\,|\,c)\right) + \underset{p(z)}{\mathbb{E}}H\left(p(c\,|\,z), q(c\,|\,z)\right) - \beta I(X;C). \tag{6}$$

If we treat the entropies of the concepts and the labels as constants, we obtain

$$\mathcal{L}_{\text{E-CIB}} = \underset{p(c)}{\mathbb{E}}H\left(p(y\,|\,c), q(y\,|\,c)\right) + \underset{p(z)}{\mathbb{E}}H\left(p(c\,|\,z), q(c\,|\,z)\right) + \beta\left(\rho - I(X;C)\right), \tag{7}$$

where $\rho$ is a constant. We denote the models that use this loss as $\text{IB}_E$ since it relies on the estimator of the mutual information. We detail the estimator we used in our implementation in Appendix B.2.

---

[1]Note that one can obtain the same loss if the optimization problem is constrained over the concepts instead, i.e., $\max I(Z;C) + I(C;Y)$ s.t. $I(X;C) \leq I_C$. Nevertheless, we present the relation with the traditional compression for completeness.

This loss is similar to the one proposed by Kawaguchi et al. [9], $\mathcal{L}_{\text{K}} = \mathbb{E}_{p(z)} H\left(p(y \mid z), q(y \mid z)\right) + \beta(\rho - I(Z; X))$, if one extends the mutual information from the labels into the concepts in a similar way. In other words, our mutual information estimated loss (7) resembles that of Kawaguchi et al.'s [9] proposal with the corresponding conditioning changes in the labels and the concepts. Thus, it is interesting to see that other optimization approaches emerge out of this bound. We highlight that our proposal is a generalized framework that encompass a wide range of possible implementations.

Unlike $\mathcal{L}_{\text{SUB-CIB}}$ (5), which simplifies the mutual information terms into cross-entropy losses, $\mathcal{L}_{\text{E-CIB}}$ retains an explicit control over $I(X; C)$. This allows for more granular control over the information flow from inputs to concepts, leading to a tighter constraint on concept leakage. As we show in the results (Table 1), this additional control translates to improved performance in both concept and class prediction accuracy, cf. Section 4.

## 4 Experiments

We extend several CBM variants with our IB-regularizers, yielding CIBMs. The CIBMs are slight variations of the original models as they require a variational approximation in order to study and apply the proposed IB-regularizers. We train each model from scratch and compare CIBMs to their vanilla counterparts of equal capacity, measuring both class-prediction accuracy and concept leakage. Our goal is to close the accuracy gap to black-box models without sacrificing interpretability or test-time intervenability. Finally, we analyze information flows via mutual-information estimates and benchmark intervention performance.

We benchmark our approach on three datasets: CUB [21], AwA2 [22], and aPY [6]. We present all implementation details in Appendix B. For our regularizers, we evaluate their setups and select the best hyperparameters (cf. Section 4.6). In the following experiments, we use the same hyperparameters and setup for our regularizers for fair comparisons.

### 4.1 Performance across all Datasets

We present the evaluation results across three datasets in Table 1. Our "black-box model" serves as a gold standard, representing the highest possible class accuracy achievable by a CBM model with a traditional setup that does not provide explanations, i.e., trained only to predict class labels. We compare against hard (H) and soft (S) CBMs trained jointly (J) or independently (I) [8], ProbCBMs [10], intervention-aware CEM [3], and AR-CBM [8]. Our main objective is to demonstrate that our proposed regularizers (IB$_B$ and IB$_E$) maintain or improve the target prediction accuracy in comparison to their original counterparts while improving the concept prediction accuracy and reducing concept leakage. The latter is of particular importance to guarantee the explainability of the results.

Our proposed methods, IB$_B$ and IB$_E$, show an improvement over all methods regarding class prediction accuracy for the *CUB dataset*, and always show improved class prediction. These improvements

**Table 1:** Accuracy results include mean and std. over 5 runs. We report results of our proposed regularizer methods, IB$_B$ and IB$_E$, applied to different CBMs. Black-box is a gold standard for class prediction that offers no explainability over the concepts.

| Method | Concept | Class |
|---|---|---|
| **CUB** | | |
| Black-box | – | 0.919±0.002 |
| CBM (HJ) | 0.956±0.001 | 0.650±0.002 |
| CBM (HJ) + IB$_B$ | 0.955±0.001 | 0.653±0.003 |
| CBM (HJ) + IB$_E$ | 0.955±0.001 | 0.656±0.003 |
| CBM (HI) | 0.956±0.001 | 0.644±0.001 |
| CBM (HI) + IB$_B$ | 0.957±0.001 | 0.686±0.000 |
| CBM (HI) + IB$_E$ | 0.957±0.001 | 0.686±0.000 |
| CBM (SJ) | 0.956±0.001 | 0.708±0.006 |
| CBM (SJ) + IB$_B$ | 0.958±0.001 | 0.725±0.004 |
| CBM (SJ) + IB$_E$ | **0.959±0.001** | 0.729±0.003 |
| ProbCBM | 0.956±0.001 | 0.718±0.005 |
| ProbCBM + IB$_B$ | 0.957±0.001 | 0.742±0.004 |
| ProbCBM + IB$_E$ | 0.957±0.001 | 0.740±0.003 |
| CEM | 0.954±0.001 | 0.759±0.002 |
| CEM + IB$_B$ | 0.955±0.001 | 0.776±0.002 |
| CEM + IB$_E$ | 0.955±0.001 | 0.776±0.002 |
| AR-CBM | 0.956±0.002 | 0.761±0.010 |
| AR-CBM + IB$_B$ | 0.956±0.003 | **0.784±0.006** |
| AR-CBM + IB$_E$ | 0.956±0.002 | 0.783±0.005 |
| **AwA2** | | |
| Black-box | – | 0.893±0.000 |
| CBM (HJ) | 0.979±0.000 | 0.853±0.002 |
| CBM (HJ) + IB$_B$ | 0.976±0.000 | 0.850±0.003 |
| CBM (HJ) + IB$_E$ | 0.979±0.000 | 0.852±0.003 |
| CBM (HI) | 0.979±0.000 | 0.836±0.001 |
| CBM (HI) + IB$_B$ | 0.975±0.000 | 0.831±0.002 |
| CBM (HI) + IB$_E$ | 0.979±0.000 | 0.835±0.002 |
| CBM (SJ) | 0.979±0.001 | 0.876±0.001 |
| CBM (SJ) + IB$_B$ | 0.979±0.002 | **0.885±0.002** |
| CBM (SJ) + IB$_E$ | 0.979±0.001 | 0.883±0.001 |
| ProbCBM | 0.979±0.000 | 0.880±0.003 |
| ProbCBM + IB$_B$ | 0.979±0.000 | 0.883±0.001 |
| ProbCBM + IB$_E$ | 0.979±0.000 | 0.882±0.001 |
| CEM | 0.979±0.000 | 0.884±0.002 |
| CEM + IB$_B$ | 0.978±0.000 | 0.883±0.003 |
| CEM + IB$_E$ | 0.979±0.000 | 0.884±0.003 |
| AR-CBM | 0.979±0.001 | 0.884±0.006 |
| AR-CBM + IB$_B$ | 0.978±0.000 | 0.885±0.008 |
| AR-CBM + IB$_E$ | 0.979±0.000 | 0.885±0.003 |
| **aPY** | | |
| Black-box | – | 0.866±0.003 |
| CBM (SJ) | 0.967±0.000 | 0.797±0.007 |
| CBM (SJ) + IB$_B$ | 0.967±0.000 | 0.856±0.005 |
| CBM (SJ) + IB$_E$ | 0.967±0.000 | 0.856±0.004 |
| ProbCBM | 0.967±0.000 | 0.863±0.007 |
| ProbCBM + IB$_B$ | 0.967±0.000 | 0.869±0.003 |
| ProbCBM + IB$_E$ | 0.967±0.000 | 0.870±0.001 |
| CEM | 0.967±0.000 | 0.869±0.004 |
| CEM + IB$_B$ | 0.967±0.000 | 0.872±0.002 |
| CEM + IB$_E$ | 0.967±0.000 | 0.876±0.003 |
| AR-CBM | 0.967±0.000 | 0.873±0.004 |
| AR-CBM + IB$_B$ | 0.967±0.000 | 0.878±0.004 |
| AR-CBM + IB$_E$ | 0.967±0.000 | **0.878±0.002** |

**Table 2:** Concept leakage evaluation (lower is better).

| Model | Complete CS | | Selective Drop-out CS | | Random Drop-out CS | |
|---|---|---|---|---|---|---|
| | OIS | NIS | OIS | NIS | OIS | NIS |
| CBM (SJ) | $4.69 \pm 0.43$ | $66.25 \pm 2.31$ | 16.29 | 78.39 | $12.97 \pm 0.78$ | $74.19 \pm 1.04$ |
| CBM (SJ) +IB$_B$ | $2.16 \pm 0.13$ | $61.67 \pm 1.92$ | 13.09 | 73.40 | $10.59 \pm 1.48$ | $71.38 \pm 0.89$ |
| CEM | $8.74 \pm 0.30$ | $75.41 \pm 3.83$ | 20.85 | 80.19 | $18.31 \pm 0.09$ | $76.56 \pm 2.00$ |
| CEM+IB$_B$ | $6.11 \pm 0.24$ | $70.02 \pm 2.21$ | 17.22 | 76.67 | $14.10 \pm 0.42$ | $72.68 \pm 2.38$ |
| AR-CBM | $3.90 \pm 0.27$ | $62.30 \pm 1.52$ | 14.16 | 63.40 | $12.58 \pm 0.86$ | $60.86 \pm 1.32$ |
| AR-CBM+ IB$_B$ | $2.83 \pm 0.27$ | $59.87 \pm 1.52$ | 10.97 | 59.72 | $10.20 \pm 0.55$ | $56.28 \pm 1.33$ |
| ProbCBM | $4.30 \pm 0.10$ | $64.22 \pm 1.04$ | 16.01 | 76.92 | $13.81 \pm 0.21$ | $75.01 \pm 0.86$ |
| ProbCBM+ IB$_B$ | $2.53 \pm 0.46$ | $60.35 \pm 2.01$ | 13.11 | 72.86 | $10.34 \pm 0.55$ | $70.96 \pm 1.91$ |

come alongside enhanced concept accuracy (in most cases and with comparable accuracy at worst), thus, realizing the fundamental goal of our approach: to simultaneously boost performance and interpretability. As for the *AwA2 dataset*, the class accuracy gain shows less improvement than the other datasets but is nevertheless comparable to the original methods. Similarly, the concept prediction is also comparable to the unregularized models. We ascribe this to the dataset's relative simplicity, which narrows the room for enhancement. In the more varied real-world classes of the *aPY dataset*, our regularizers significantly outperforms the baseline CBMs in class accuracy. We even observed an improvement over the black-box model while providing interpretability comparable to the original models, which is paramount in real-world applications where explanations are necessary.

The rise in class and concept accuracy relative to existing methods highlights the advantages of our mutual information regularization. This approach helps stop concept leakage and ensures that concepts are both informative and closely tied to the final prediction, see Section 4.2 for details. This finding is consistent with our theoretical framework, which advocates that controlling the information flow between inputs and concepts through the Information Bottleneck can yield more interpretable and significantly meaningful concepts without compromising performance, see Section 4.5 for more details.

## 4.2 Concept Leakage

Concept leakage occurs when spurious or task-irrelevant information contaminates concept activations eroding both interpretability and the power of test-time interventions [13, 14]. Espinosa Zarlenga et al. [4] proposed Oracle Impurity Score (OIS) and the Niche Impurity Score (NIS) to quantify impurities localized within individual and distributed across the set of learned concepts, respectively. We use these metrics under three scenarios: (i) a complete concept set, (ii) selective dropout where we remove the most predictive half of concepts, and (iii) random dropout of half of concepts (for control). We highlight that the selective dropout setting has only one possible configuration, thus, we do not report standard deviation on it. We chose dropout scenarios because omitting relevant concepts can dramatically increase concept leakage [8]. Table 2 reports these results. Crucially, our IB-regularizers significantly slash leakage across all scenarios, achieving the lowest OIS and NIS even under heavy concept removal. These results confirm that imposing an Information Bottleneck on concepts reduces concept leakage and mitigates spurious encoding.

## 4.3 Interventions

A key advantage of CBMs is their ability to perform *test-time interventions*, allowing users to correct predicted concepts and improve the model's final decisions. To demonstrate test-time intervention performance of CIBMs we simulate interventions by replacing predicted concepts with their ground truth values. Following prior work, we intervene on *groups of concepts* rather than individual concepts, leveraging this strategy to assess how cumulative corrections impact class prediction performance [10, 12]. We, then, plot the prediction performance improvement against number of concept groups intervened. The resulting curve is denoted as the interventions curve. We implement a random strategy to choose a set of concept groups to intervene on. More specifically, concept groups are randomly selected for intervention, and results are averaged over five runs to account for variability.

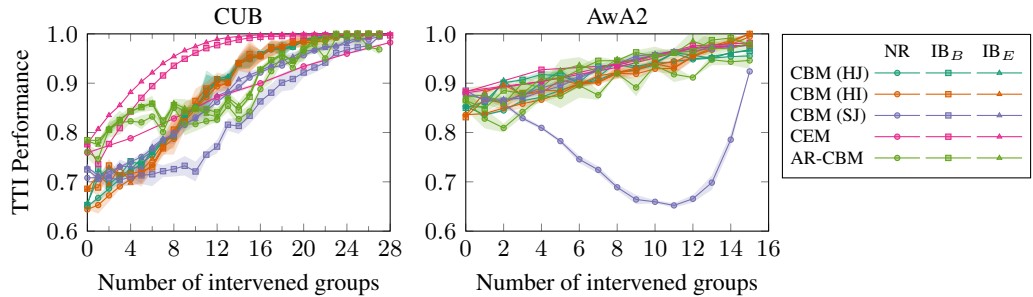

**Figure 3:** Change in target prediction accuracy after intervening on concept groups following the random strategy as described in Section 4.3. (TTI stands for Test-Time Interventions, and NR for non-regularized.) We show expanded plots in Fig. C.1.

**Table 3:** Change in interventions performance with concept set corruption for CBM (SJ) and its regularized versions with our proposed methods. We show the disaggregated plots in Fig. C.2.

| | CUB | | | | | |
|---|---|---|---|---|---|---|
| | AUC | | | NAUC | | |
| Corrupt | CBM | +IB$_B$ | +IB$_E$ | CBM | +IB$_B$ | +IB$_E$ |
| 0 | 54.374 | 65.644 | 64.634 | 0.001260 | 0.001481 | 0.001432 |
| 4 | 53.135 | 64.519 | 63.464 | 0.001198 | 0.001525 | 0.001487 |
| 8 | 51.291 | 53.135 | 60.202 | 0.001166 | 0.001198 | 0.001444 |
| 16 | 50.694 | 60.240 | 59.424 | 0.001068 | 0.001388 | 0.001349 |
| 32 | 46.101 | 52.956 | 51.258 | 0.000863 | 0.001298 | 0.001231 |
| 64 | 32.069 | 30.582 | 29.271 | -0.000339 | 0.000571 | 0.000504 |
| | AwA2 | | | | | |
| | AUC | | | NAUC | | |
| Corrupt | CBM | +IB$_B$ | +IB$_E$ | CBM | +IB$_B$ | +IB$_E$ |
| No | 84.753 | 91.573 | 92.225 | 0.002808 | 0.005350 | 0.006250 |
| Yes | 83.985 | 90.631 | 90.879 | 0.004484 | 0.005218 | 0.006474 |

Figure 3 shows that IB-regularized CBMs deliver a monotonic rise in accuracy as each additional concept group is corrected—clear evidence they truly leverage accurate concept signals with minimal leakage. This smooth ascent underscores how our bottleneck penalty sharpens the model's debuggability, ensuring every intervention yields a consistent performance boost. In contrast, soft-joint CBMs suffer pronounced mid-sequence dips—likely a symptom of their leaky representations undermining reliability under random group corrections.

Hard CBMs—with their binary concept slots—can eventually attain high accuracy under large-scale interventions (owing to their inherently low leakage), but they start well below CIBMs and climb more sluggishly when only a few concepts are corrected—especially on coarser datasets like AwA2. In contrast, our IB-regularized models blend low-leakage encodings with adaptive flexibility, producing smooth, steady gains and outperforming every CBM variant in both intervention curves and overall accuracy (Table 1). For full setup details, see Appendix F. Interestingly, current models, such as CEM and AR-CBM, benefit the most from our regularization showing a significant improvement in both data sets.

## 4.4 Concept Set Goodness Measure

In CBMs, the quality of the concept set is crucial for accurate downstream task predictions. However, there is a lack of effective metrics to reliably assess concept set goodness. Existing metrics, such as the Concept Alignment Score, proposed by Espinosa Zarlenga et al. [3], evaluate whether the model has captured meaningful concept representations but do not explicitly measure how well these concepts improve downstream task performance during interventions. Moreover, this metric is tuned for CEM and do not extend beyond it.

Similar to previous methods that rely on area under the curve for the interventions [5, 18], we measure and compare the concept quality in CIBMs using the following metrics: area under interventions curve, and the area under curve of relative improvements. Denote by $\mathcal{I}(x)$ the model's performance for $x$ concept groups used in the intervention. Then the Test-Time Interventions (TTI) accuracy is

$$\text{AUC}_{\text{TTI}} = \frac{1}{n} \sum_{i=1}^{n} \mathcal{I}(i), \tag{8}$$

and the normalized version of the TTI accuracy is

$$\text{NAUC}_{\text{TTI}} = \frac{1}{n} \sum_{i=1}^{n} \left( \mathcal{I}(i) - \mathcal{I}(i-1) \right). \tag{9}$$

The idea behind these measures is simple: if a concept set is of high quality, the task accuracy will steadily approach $100\%$ as more concept groups are intervened upon, resulting in a large area under the curve. Conversely, if the concept set is incomplete or noisy, performance gains will be limited, even with multiple interventions, which can indicate concept leakage.

The latter expression (9) could be simplified to just scaled difference between a model with full concept set used for interventions and performance of a model with no interventions, however, the meaning it has is how much does the performance change per one group added to the interventions pool. To test this, we generate corrupted concept sets by replacing selected concepts with noisy ones. Importantly, we maintain the original groupings of concepts.

Table 3 shows the results of our metrics. We also show the commonly reported disaggregated plots in Fig. C.2. The number in the "corrupt" column denotes the number of concepts replaced with random ones for CUB, and for AwA2 "No" denotes a clear concept set and "Yes" denotes a concept set with one concept changed to corrupt. As expected, performance drops with corrupt concepts, since they contain no useful information for the target task. One consequence of our training is that if one has two concept annotations for some dataset, then it is possible to use CIBMs performance to determine which concept set is better.

Our results demonstrate that regularizing with $\text{IB}_E$ is more sensitive to concept quality compared to vanilla CBM, making it a better indicator of concept set reliability. Negative values in normalized intervention AUC indicate possible concept leakage.

### 4.5 Information Plane Dynamics in CBMs and CIBMs

To further evaluate the proposed regularizers, we examined the information plane dynamics of CBM, CEM, and AR-CBM, as shown in Fig. E.1. In general, we expected to observe higher mutual information between the concepts and the labels, $I(C; Y)$, and between the latents and the concepts, $I(Z; C)$, while expecting lower mutual information between the data and the concepts, $I(X; C)$, and between the data and the latents, $I(X; Z)$. We clearly observed this behavior when applying our $\text{IB}_E$ to CEM, and to a lesser degree with $\text{IB}_B$. This pattern was also evident in AR-CBMs, although with more noise. However, in certain cases, this pattern deviated. More specifically, we found that CBMs exhibit greater compression with respect to the data compared to their regularized counterparts. Nevertheless, our CIBMs demonstrate greater expressiveness due to their higher mutual information with respect to the labels, $Y$.

We think that vanilla CBMs "over-compress" their internal representations—shrinking $I(X; C)$ and $I(X; Z)$ so aggressively that they discard useful, task-relevant features. This indiscriminate bottleneck explains their lower end-to-end accuracy (Table 1) and higher concept leakage (Table 2). By contrast, our CIBMs apply a *structured* Information Bottleneck: they retain all the signal that drives $Y$ (higher $I(C; Y)$) while shedding only the noise (lower $I(X; C)$), which both boosts predictive performance and cuts leakage. In other words, achieving **expressiveness first**—then **selective compression**—yields representations that are both robust and interpretable. Appendix E presents detailed information-plane trajectories, and our findings echo recent theory on IB in deep nets, which warns against blind compression in favor of task-guided pruning [9].

Overall, we have found that pursuing compression alone is not the solution for obtaining more robust representations. Instead, we see that achieving more expressive representations (i.e., higher mutual information with respect to the labels) followed by compression (i.e., lower mutual information with respect to the data) helps reduce the gaps in predictive tasks (see Table 1) as well as in leakage (see

Table 2). However, due to the requirements for expressiveness, the CIBMs do not compress as much, since they must retain some useful information. Our findings align with recent theoretical insights on the Information Bottleneck principle in deep learning [9], which emphasize that indiscriminately minimizing the mutual information between the data and the latent representations, $I(X; Z)$, does not guarantee expressive or generalizable representations. Effective models must selectively compress task-irrelevant information while retaining essential features for decision-making.

## 4.6 Evaluation of our Regularizers' Hyperparameters

We evaluated the hyperparameters of our proposed regularizers on a CBM (SJ) to select the values that we used for all other experiments. We evaluated our regularizers in a single model to find the best setup due to computational constraints. We compare the performance of $IB_B$ and $IB_E$ on concept and class prediction accuracy for the CUB dataset (using $\beta = 0.5$) and report the results in Table B.1. As shown, $IB_E$, which retains an explicit mutual information term $I(X; C)$, outperforms $IB_B$ when trained in a fair setup (vanilla) in both metrics. We found that the lack of performance of the vanilla $IB_B$ regularizer comes from instabilities during training in the latent representations encoder $p(z \mid x)$. We hypothesize that the gradient from the $H(C)$ in the loss (5) damages the feature encoder $p(z \mid x)$ since the entropy is computed w.r.t. the generative concepts $p(c)$ instead of the variational approximated ones $q(c)$. To alleviate this problem, we experimented gradient clipping as well as stopping the gradient from $H(C)$ into the encoder. We found that the latter performs on par with $IB_E$. In the experiments, we use $IB_B$ with stop gradient on it. Overall, $IB_E$'s more granular control over information flow limits concept leakage, results in better accuracies for concepts and labels in comparison to the baselines (cf. Table 1) without changes to its training framework. We also evaluate two different values (0.25 and 0.5) for the $\beta$ constant that controls the mutual information between the data and the concepts. We show these results in Table B.2. Since we obtained inconclusive results, we selected $\beta = 0.5$ for following experiments.

These results supports our earlier discussion that the direct estimation of $I(X; C)$ leads to more effective use of concepts in downstream tasks without further changes to the training regime. Nevertheless, with a correctly regularized feature encoder $p(z \mid x)$, a simple estimation in $IB_B$ can achieve similar levels of information gain and accuracy.

## 5 Limitations

Our reliance on variational MI estimation can introduce bias and depend sensitively on the choice of approximating distributions and estimators used (as shown in our results for our two variations of regularizers). In general, like all CBMs, CIBMs assume reliable, comprehensive concept annotations—performance and leakage gains may diminish if concept labels are noisy, incomplete, or inconsistency defined, though our results have demonstrated that CIBMs are more robust to incomplete concepts as compared to their corresponding state of the art variants.

## 6 Conclusion

We present *Concepts' Information Bottleneck Models* (CIBMs), a first-principled fusion of Information Bottleneck theory and Concept Bottleneck Models that both explains CBMs' failure modes and prescribes their cure. By penalizing $I(X; C)$ while preserving $I(C; Y)$, Concept Information Bottleneck reveals why vanilla CBMs over-compress and leak spurious signals—and how a surgical, task-guided compression can retain exactly what matters. We validate CIBMs across six CBM families (hard/soft, joint/independent, ProbCBM, CEM, and AR-CBM) on three benchmarks (CUB, AwA2, and aPY), employing concept accuracy, class accuracy, Oracle and Niche Impurity (OIS and NIS), and intervention metrics ($AUC_{TTI}$, $NAUC_{TTI}$). The result is uniformly higher class accuracy, dramatically reduced concept leakage, and equal or better concept-prediction performance—closing much of the CBM-black-box gap. Crucially, our findings show that: (a) **simple, selective compression** can unlock robust, interpretable concept representations; and (b) that leakage undermines the *use* of concepts far more than their *detection*, explaining why near-perfect concept predictors can still yield subpar end-to-end performance.

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

# A  Detailed Derivation of CIB

In this section we present the detailed derivations to obtained the results described in Section 3.1.

We can re-write the upper bound of the concepts' information bottleneck as

$$\mathcal{L}_{\text{UB-CIB}} = H(Y) + (1 - \beta)H(C) - H(Y \mid C) - H(C \mid Z) - \beta H(C \mid X) \tag{A.1}$$

to work with the entropies instead. To find a more suitable form to tackle this bound, we consider an approximation of the predictors for the labels and the concepts, $q(y \mid c)$ and $q(c \mid z)$, based on two variational distributions that will be implemented through neural networks—cf. Fig. 2. Consider, on one hand,

$$H(Y \mid C) = \iint dy \, dc \, p(y, c) \log p(y \mid c), \tag{A.2a}$$

$$= \iint dy \, dc \, p(y, c) \log \left[ p(y \mid c) \frac{q(y \mid c)}{q(y \mid c)} \right], \tag{A.2b}$$

$$= \iint dy \, dc \, p(y \mid c) p(c) \left[ \log \frac{p(y \mid c)}{q(y \mid c)} + \log q(y \mid c) \right], \tag{A.2c}$$

$$= \int dc\, p(c) \int dy\, p(y \mid c) \left[ \log \frac{p(y \mid c)}{q(y \mid c)} + \log q(y \mid c) \right], \tag{A.2d}$$

$$= \underset{p(c)}{\mathbb{E}} \left[ \mathrm{KL}\big(p(y \mid c) \,\|\, q(y \mid c)\big) - H\left(p(y \mid c), q(y \mid c)\right) \right]. \tag{A.2e}$$

We introduce the variational distribution $q(y \mid c)$ to obtain the cross-entropy w.r.t. the ground truth and this results on an additional term to make the variational distribution close to the prior. In other words, we can interpret the conditional entropy of the labels w.r.t. the concepts as an optimization of the variational distribution $q(y \mid c)$ with the true conditional of the labels given the concepts $p(y \mid c)$ through a Kullback-Leibler divergence (KL) and the cross-entropy between them. This last cross-entropy can be interpreted as the traditional prediction loss of the true labels and the predicted ones. Similarly,

$$H(C \mid Z) = \iint dc\, dz\, p(c, z) \log p(c \mid z), \tag{A.3a}$$

$$= \iint dc\, dz\, p(c, z) \log \left[ p(c \mid z) \frac{q(c \mid z)}{q(c \mid z)} \right], \tag{A.3b}$$

$$= \iint dc\, dz\, p(c \mid z) p(z) \left[ \log \frac{p(c \mid z)}{q(c \mid z)} + \log q(c \mid z) \right], \tag{A.3c}$$

$$= \int dz\, p(z) \int dc\, p(c \mid z) \left[ \log \frac{p(c \mid z)}{q(c \mid z)} + \log q(c \mid z) \right], \tag{A.3d}$$

$$= \underset{p(z)}{\mathbb{E}} \left[ \mathrm{KL}\big(p(c \mid z) \,\|\, q(c \mid z)\big) - H\left(p(c \mid z), q(c \mid z)\right) \right], \tag{A.3e}$$

were $q(c \mid z)$ is a variational distribution that predicts the concepts given the latent representations. This decomposition of the conditional entropy of the concepts given the representations follows the same principles as the conditional of the labels given the concepts (A.2). On the other hand, the conditional entropy of the concepts w.r.t. the data is bounded due to the marginalization of the latent representations on their dependency. That is,

$$H(C \mid X) = \iint dc\, dx\, p(c, x) \log p(c \mid x), \tag{A.4a}$$

$$= \iint dc\, dx\, p(c, x) \log \int dz\, p(c, z \mid x), \tag{A.4b}$$

$$= \iint dc\, dx\, p(c, x) \log \int dz\, p(c \mid z) p(z \mid x), \tag{A.4c}$$

$$\leq \iint dc\, dx\, p(c, x) \int dz\, p(z \mid x) \log p(c \mid z), \tag{A.4d}$$

$$= \iiint dc\, dz\, dx\, p(c, z, x) \int dz\, p(z \mid x) \log p(c \mid z), \tag{A.4e}$$

$$= \iiint dc\, dz\, dx\, p(c \mid z) p(z \mid x) p(x) \int dz\, p(z \mid x) \log p(c \mid z), \tag{A.4f}$$

$$= \int dx\, p(x) \iiint dc\, dz^2\, p(c \mid z) p(z \mid x)^2 \log p(c \mid z), \tag{A.4g}$$

$$= \int dx\, p(x) \iiint dc\, dz^2\, p(c \mid z) p(z \mid x)^2 \log \left[ p(c \mid z) \frac{q(c \mid z)}{q(c \mid z)} \right], \tag{A.4h}$$

$$= \int dx\, p(x) \iint dz^2\, p(z \mid x)^2 \int dc\, p(c \mid z) \log \left[ p(c \mid z) \frac{q(c \mid z)}{q(c \mid z)} \right], \tag{A.4i}$$

$$= \underset{p(x)}{\mathbb{E}}\, \underset{p(z \mid x)}{\mathbb{E}} \int dc\, p(c \mid z) \log \left[ p(c \mid z) \frac{q(c \mid z)}{q(c \mid z)} \right], \tag{A.4j}$$

$$= \underset{p(z \mid x) p(x)}{\mathbb{E}} \int dc\, p(c \mid z) \left[ \log \frac{p(c \mid z)}{q(c \mid z)} + \log q(c \mid z) \right], \tag{A.4k}$$

$$= \underset{p(z \mid x) p(x)}{\mathbb{E}} \left[ \mathrm{KL}\big(p(c \mid z) \,\|\, q(c \mid z)\big) - H\left(p(c \mid z), q(c \mid z)\right) \right], \tag{A.4l}$$

where the bound comes from applying the Jensen's inequality. Thus, the upper bound to the concept bottleneck loss (2), given that we remove the KLs constraints, due to their positivity, from the conditional entropies (A.2), (A.3) and (A.4) is

$$\mathcal{L}_{\text{UB-CIB}} \leq H(Y) + (1-\beta)H(C) + \underset{p(c)}{\mathbb{E}} H\left(p(y \mid c), q(y \mid c)\right) + (1+\beta)\underset{p(z)}{\mathbb{E}} H\left(p(c \mid z), q(c \mid z)\right). \quad \text{(A.5)}$$

The bound gap can be further reduced by dropping the entropy of the labels as

$$\mathcal{L}_{\text{UB-CIB}} \leq (1-\beta)H(C) + \underset{p(c)}{\mathbb{E}} H\left(p(y \mid c), q(y \mid c)\right) + (1+\beta)\underset{p(z)}{\mathbb{E}} H\left(p(c \mid z), q(c \mid z)\right), \quad \text{(A.6)}$$

$$= \mathcal{L}_{\text{SUB-CIB}}. \quad \text{(A.7)}$$

In other words, we can maximize the concepts' information bottleneck by minimizing the cross entropies of the predictive variables, $y$ and $c$, and their corresponding ground truths and by adjusting the entropy of the concepts.

# B    Implementation Details

## B.1    Details on the Models

To regularize existing models, we take the layer in their architecture that outputs the latent representation and insert a variational reparametrization to it. That is, we insert two heads that output the mean and standard deviation for our variational approximation based on the architecture, and sample the latents from them. In a nutshell for these heads, we add on top of the model's embedding layer (the bottleneck of the model) two 1-layer MLP (i.e., our heads), for mean and standard deviation using the reparametrization trick in the variational approximation $q(c \mid z)$, each of dimensionality 112—the number of concepts left after filtration identical to one done in Koh et al.'s [12] work. For CEMs, we introduce variational approximation for every concept embedding projection. We obtain concept logits as $C = \text{pred}_\mu(x) + \text{pred}_\sigma(x) \cdot \epsilon$, where $\epsilon$ is a random standard Gaussian noise. On top of concepts logits, we stack label predictor $q(y \mid c)$ (also 1-layer MLP). All activations between the layers are ReLU. For the CUB dataset, we choose for each original CBM-like model the respective image encoding backbone as image embedder $p(z \mid x)$. For AwA2 and aPY the only difference is that we use on pre-computed embeddings from ResNet18 without training the backbone.

For CEM [3] there are basically two training options: intervention-aware and basic. In the latter, the model just optimizes two CE objectives. We implemented and trained the intervention-aware setup on CUB, AwA2, and aPY. Then, we measured the interventions performance.

Our accuracies coincided with those reported by Espinosa Zarlenga et al. [3] in their paper on CUB dataset. And intervention performance of this intervention-unaware model variant matched the reported behavior from the authors (i.e., no gain from interventions).

## B.2    Estimators Details

**Mutual Information Estimator.** Before each gradient update, we compute cross-entropies over the current batch $B_c$, and then randomly sample batch $B_c'$ from the training dataset to estimate $I(X; C)$ on this batch.

Our mutual information estimator follows Kawaguchi et al.'s [9] work. We rely on the fact that concepts logits have Gaussian distribution for estimation of $\log p(c \mid x)$. And then, we use the random samples $B_c'$ to approximate the marginal of the concepts $\log p(c)$. The mutual information $I(C; X)$ is then a Monte-Carlo estimate of $\log p(c \mid x) - \log p(c)$.

**Entropy Estimator.** Since concepts $C$ are distributed normally, we use $H(C) = \frac{D}{2}(1 + \log(2\pi)) + \frac{1}{2}\log|\Sigma|$. For simplicity (since the number of concepts $D$ is constant throughout the training and inference) we use $\hat{H}(C) = \frac{1}{2}\log|\Sigma| = \sum \log(\sigma_i)$ since $\Sigma$ is a diagonal matrix in our setup.

## B.3    Training Parameters

We explained the hyperparameter selection in Section 4.6. We experimented with different setups to find the best configuration. We show these results in Tables B.1 and B.2. The other training

**Table B.1:** Accuracies of CBM (SJ) with our proposed regularizers, $IB_B$ and $IB_E$, on CUB dataset (avg. 3 runs).

| Method | Concept | Class |
|---|---|---|
| $IB_B$ (vanilla) | 0.934 | 0.608 |
|     (clip_norm = 1.0) | 0.947 | 0.660 |
|     (clip_norm = 0.1) | 0.947 | 0.646 |
|     (stop grad. from $H(C)$ into $p(z \mid x)$) | **0.959** | 0.726 |
| $IB_E$ | **0.959** | **0.729** |

**Table B.2:** Evaluation of CBM (SJ) with the proposed regularizers on three datasets with two different values of $\beta$.

| | $\beta$ | 0.25 | | 0.50 | |
|---|---|---|---|---|---|
| Dataset | Method | Concept | Class | Concept | Class |
| CUB | $IB_B$ | 0.958±0.001 | 0.726±0.003 | 0.958±0.001 | 0.725±0.004 |
| | $IB_E$ | 0.958±0.001 | 0.728±0.005 | 0.959±0.001 | 0.729±0.003 |
| AwA2 | $IB_B$ | 0.980±0.000 | 0.886±0.002 | 0.979±0.000 | 0.885±0.002 |
| | $IB_E$ | 0.980±0.000 | 0.885±0.001 | 0.979±0.000 | 0.883±0.001 |
| aPY | $IB_B$ | 0.967±0.000 | 0.850±0.006 | 0.967±0.000 | 0.856±0.005 |
| | $IB_E$ | 0.967±0.000 | 0.858±0.004 | 0.967±0.000 | 0.856±0.004 |

parameters for the models are as follows. We set batch size to 128 and number of samples for MI estimation to 64. For all experiments we used Adam [11] optimizer with $lr = 0.003$ and $wd = 0.001$. We experimented with gradient clipping, but it led to either slow or divergent training, so we are not clipping the gradients in any of the experiments.

## B.4 Datasets

We benchmark our approach on 3 datasets: CUB [21], AwA2 [22], and aPY [6]. While CUB is a recognized dataset for comparing concept-based approaches [3, 10, 12], we add the other two datasets for additional evaluations and analysis.

**CUB.** Caltech-UCSD Birds dataset [21] is a dataset of birds images totaling in 11788 samples for 200 species. Following Koh et al.'s [12] work, for reproducibility, we reduce instance-level concept annotations to class-level ones with majority voting. We then keep only the concept that are annotated as present in 10 classes at least after the described voting, resulting in 112 concepts instead of 312. We also employ train/val/test splits provided by Koh et al. [12], operating with 4796 train images, 1198 val images and 5794 test images. To diversify training data, we augment the images with color jittering and horizontal flip, and resize the images to $299 \times 299$ pixels for the InceptionV3 backbone. Concept groups are obtained by common prefix clustering.

**AwA2.** Animals with attributes dataset [22] is a dataset of 37322 images of 50 animal species. For the concepts set, we follow Kim et al.'s [10] work and keep only the 45 concepts which could be observed on the image. We use ResNet18 embeddings provided by the dataset authors and train FCN on top of them. No additional augmentations are applied to those embeddings.

**aPY.** This is a dataset [6] of 32 diverse real-world classes we used for proof of concept. We split the dataset into 7362 train, 3068 validation and 4909 test samples stratified on target labels. We train FCN on top of ResNet18 embeddings of input images provided by the dataset authors [22]. No additional augmentations are applied to those embeddings.

## B.5 Details on Experiments

The image embedder backbone is only trained for CUB dataset [21], and for AwA2 [22] and aPY [6] we use pre-computed image embeddings. The ground truth concept labels are binary across all dataset, but concepts predictions passed to label classifier are non-binary: we are training only (and comparing only against) models using soft concepts for class prediction.

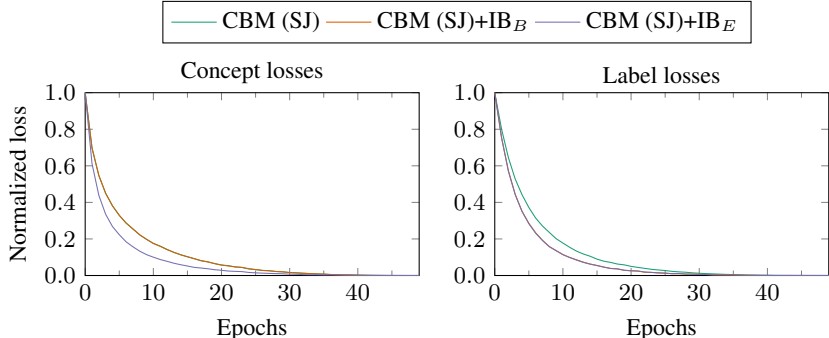

**Figure B.1:** Losses on the validation set of CUB for CBM (SJ) and its variants regularized with our proposed methods.

When training models with $IB_B$, we used the $\mathcal{L}_{\text{SUB-CIB}}$ (5) for better performance. We backpropagate the gradients from the cross-entropies over concepts and labels through the entire network—both backbone $q(c \mid z)$ and MLPs on top of the encoder $q(y \mid c)$. For $H(C)$, however, the situation is different: gradients from this part of the loss function are propagated only through the MLPs, $q(c \mid z)$ and $q(y \mid c)$, but not the image embedder backbone $p(z \mid x)$. We found that such (partial) "freezing" of the encoder with respect to $H(C)$ constraint dramatically improves the quality of both concepts and labels prediction. While we do not have access to the ground truth probability distribution for the concepts $p(c \mid z)$, we have access to the ground truth concept labels. Our implementation uses the a supervised cross-entropy using the ground truth labels. The concepts' predictor can be seens as a multi-label task classifier. In practice, we compute $C$ logits, then, we compute binary cross-entropy (BCE) for each of these logits with binary labels. Finally, we backpropagate them through the means of BCEs.

We show the normalized loss function values on the validation set of CUB in Fig. B.1 to show the convergence of CIBMs in comparison to CBM (SJ). Note that visually the concept losses on between CBM (SJ) and its variant regularized with $IB_E$ and the label losses between CIBMs are similar, but they differ slightly.

## C Extended Results on Interventions

In Fig. C.1, we show the plots of Fig. 3 separated and grouped by the type of method and dataset in order to better visualize the trends. We highlight that the fewer points in the results for CEM follows the results from Espinosa Zarlenga et al. [3].

In Fig. C.2, we show additional results about the aggregated interventions that we dicussed in Section 4.4 and that we showed in Table 3. We plot the interventions in the traditional way by showing the intervened groups and the TTI performance for six different corruption settings.

## D Extended Results on Concept Leakage

### D.1 OIS and NIS metrics

The Oracle Impurity Score (OIS) [4] quantifies impurities localized within individual concept representations. Given a concept encoder $g : X \mapsto \hat{C} \subseteq \mathbb{R}^{d \times k}$, test samples $\Gamma_X$, and their concept annotations $\Gamma$, OIS is defined as:

$$\text{OIS}(g, \Gamma_X, \Gamma) := \frac{2\|\pi(g(\Gamma_X), \Gamma) - \pi(\Gamma, \Gamma)\|_F}{k} \tag{D.1}$$

where $\pi(\hat{\Gamma}, \Gamma)$ is a purity matrix whose entries $\pi(\hat{\Gamma}, \Gamma)_{(i,j)}$ contain the AUC-ROC score when predicting the ground truth value of concept $j$ given the $i$-th concept representation. The normalization ensures OIS ranges in $[0, 1]$, with 0 indicating perfect alignment between the predictive capacity of learned and ground truth concepts.

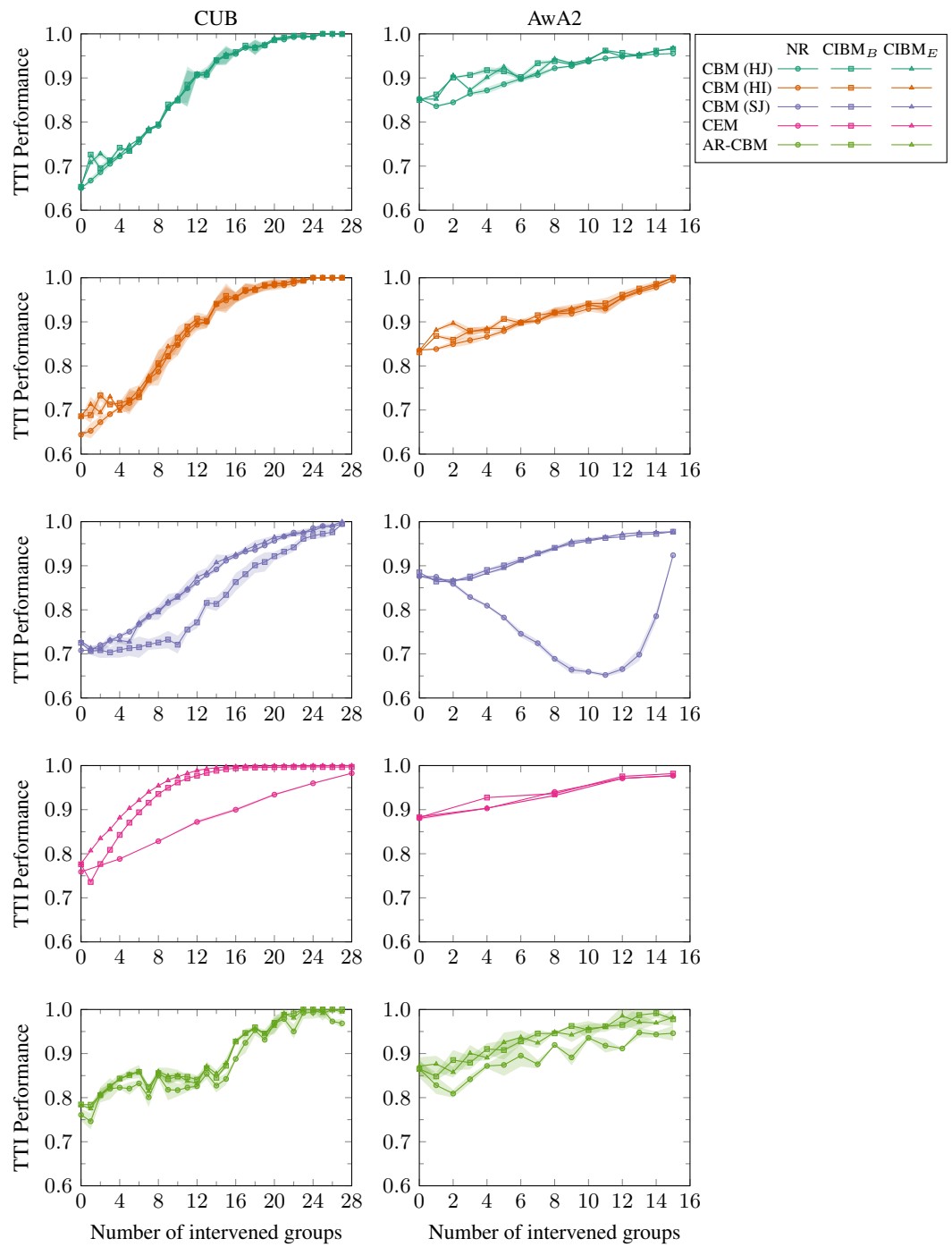

**Figure C.1:** Expanded results from Fig. 3. Change in target prediction accuracy after intervening on concept groups following the random strategy as described in Section 4.3. (TTI stands for Test-Time Interventions and NR for non-regularized.)

The Niche Impurity Score (NIS) [4] captures impurities distributed across multiple concept representations. For each concept $j$, a concept niche $N_j(\nu, \beta)$ is defined as the set of concept indices whose representations are highly entangled with concept $j$ according to a concept nicher function $\nu$ and threshold $\beta$. The Niche Impurity (NI) for concept $i$ measures how predictable this concept is from

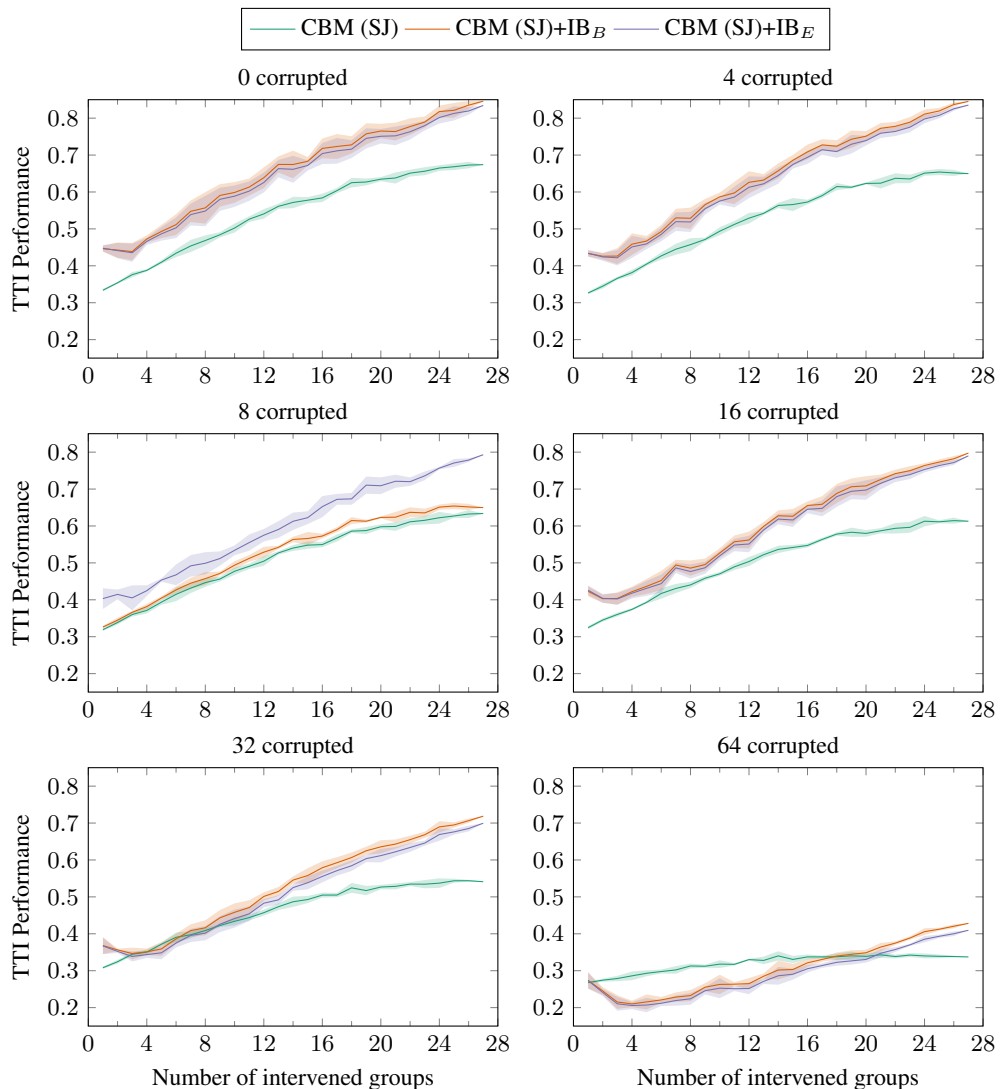

**Figure C.2:** Change in target prediction accuracy for different number of corrupted concepts. These are the expanded results of Table 3. (TTI stands for Test-Time Interventions.)

representations outside its niche:

$$\text{NI}_i(f, \nu, \beta) := \text{AUC-ROC}(\{(f|_{\neg N_i(\nu,\beta)}(\hat{c}^{(l)}_{(:, \neg N_i(\nu,\beta))}), c^{(l)}_i)\}^n_{l=1}). \tag{D.2}$$

The overall NIS is then calculated by integrating NIs across all concepts and threshold values:

$$\text{NIS}(f, \nu) := \int_0^1 \left( \sum_{i=1}^k \frac{\text{NI}_i(f, \nu, \beta)}{k} \right) d\beta. \tag{D.3}$$

A NIS of $0.5$ indicates random performance (no impurity), while a NIS of $1$ suggests that concept information is dispersed across multiple representations. Together, these metrics effectively evaluate concept quality without making unrealistic assumptions about concept independence or representation dimensionality.

## D.2 Concept sets reduction

We employed two different algorithms to cut the concepts set to half the size: selective (information-based) and random dropout. In the former, we computed $\mathbb{E}[I(Y; C_i)]$ for all concept groups on a

subsample of the training set. Then we dropped out the concepts groups with the highest mutual information—that is, we made the "fair" (leakage-free) learning as unprofitable and hard as possible. On the other hand, the random dropout selects half of the concepts at random and drops the rest.

# E    Information Plane Dynamics

We analyze the flow of information between inputs, $X$, latents, $Z$, concepts, $C$, and labels, $Y$, and present them in Fig. E.1. The objective of the information plane is to show the mutual information on the model variables after training. In particular, we expect to see a model with high $I(Z;C)$ and $I(C;Y)$ such that the corresponding variables are dependent on each other (maximally expressive), and simultaneously, low $I(X;C)$ and $I(X;Z)$ to show that the corresponding variables are maximally compressive. However, the compression of the variables alone, minimal $I(X;C)$ or $I(X;Z)$, does not guarantee that the important parts of the variables are being compressed and retained. Thus, we show the other experiments to complement this analysis.

CEM has a lower mutual information between the inputs and the latent and concept representations, $I(X;Z)$ and $I(X;C)$, than CBM (SJ). Interestingly, our regularizers reduce these mutual information while maintaining the mutual information w.r.t. the target, $I(C;Y)$ and $I(Z;C)$. However, for CBM (SJ), our methods increase the mutual information w.r.t. the data. This behavior may reflect the fact that CIBMs are optimized to retain task-relevant information while removing irrelevant or redundant information but not necessarily compressing as much—reflected in the higher $I(X;C)$ and $I(X;Z)$. Nevertheless, lower mutual information $I(X;C)$ and $I(X;Z)$ in CBMs does not necessarily indicate better compression given its lower predictive accuracy. Instead, it may reflect a failure to capture meaningful input features, resulting in noisier or less predictive concepts. Moreover, we note that the plots in Fig. E.1(f) for IB$_B$ and IB$_E$ look similar but they differ in hundredths.

For AR-CBM, the information flow is more noisy. Despite the noise, we can observe that CIBMs obtain higher mutual information w.r.t. the labels than their vanilla counterpart. While the compression w.r.t. the data is not as evident, the final mutual information w.r.t. the data is closer between the original method and its regularized versions. Nevertheless, we still observed better predictive performance (cf. Table 1). Thus, we hypothesize that the regularizer is increasing the expressiveness of the representations with a trade-off of the compression as observed with the CBMs but not as apparent. On the other hand, the CIBMs obtain better compression-expression patterns for the latent representations, see Table E.1(d).

To demonstrate the effects of the compression patterns, we evaluate the alignment between representations and the target $I(C;Y)$ and show that CIBMs consistently outperform CBMs, and, while noisy, they show improvements over CEM, indicating that the retained information is both relevant and predictive—cf. Section 4.1. Additionally, CIBMs achieve better interpretability and concept quality, reinforcing that the higher mutual information is a reflection of meaningful expressiveness rather than leakage—cf. Section 4.3. This is further supported by the proposed intervention-based metrics (AUC$_{TTI}$ and NAUC$_{TTI}$) which highlight the importance of retaining task-relevant information in the concepts $C$. While CBMs exhibit lower mutual information between inputs and representations in contrast to the regularized versions, $I(X;C)$ and $I(X;Z)$, their poorer performance on these metrics, particularly under concept corruption, suggests that this lower information content stems from a failure to capture sufficient relevant features. By contrast, the higher $I(X;C)$ and $I(X;Z)$ in our CIBMs reflect the retention of meaningful pieces that contribute to better concept quality and downstream task performance. These findings demonstrate that reducing concept leakage requires selectively preserving relevant information rather than minimizing mutual information indiscriminately.

Our findings align with recent theoretical insights on the Information Bottleneck principle in deep learning [9], which emphasize that indiscriminately minimizing the mutual information between the data and the latent representations, $I(X;Z)$, does not guarantee expressive or generalizable representations. Instead, effective models must selectively compress task-irrelevant information while retaining essential features for decision-making. Our results (cf. Table 1 and Fig. E.1) support this trade-off by demonstrating that CBMs, despite lower $I(X;C)$ and $I(X;Z)$, do not necessarily achieve superior concept representations or intervention efficacy in comparison to their IB regularized counterparts. In contrast, our IB-based CBMs, which balance information retention and compression, lead to improved alignment between concepts and final predictions, reinforcing the importance of controlled, task-relevant compression rather than absolute mutual information minimization.

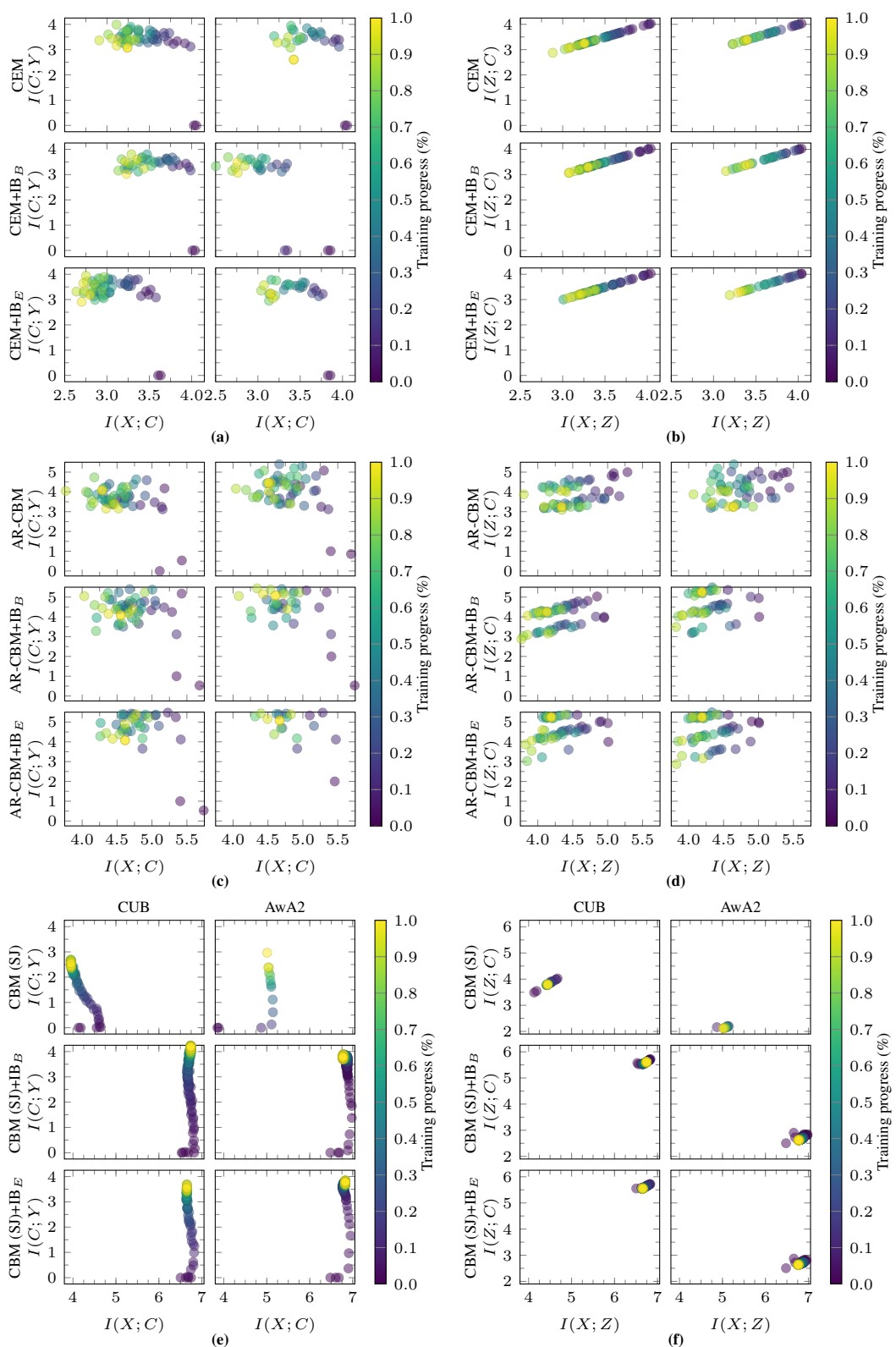

**Figure E.1:** Information plane dynamics (in nats) for (a,b) CEM, (c,d) AR-CBM, (e, f) CBM (SJ) and our proposed methods, IB$_B$ and IB$_E$. Warmer colors denote later steps in training. We show the information plane of (a, c, e) the variables $X$, $C$, and $Y$; and (b, d, f) the variables $X$, $Z$, and $C$.

# F Discussion about CBMs setups

Hard CBMs use hard concept representations, meaning that instead of producing a probabilistic output (as in soft concepts in soft CBM), each concept prediction is treated as a discrete binary or categorical value. These hard predictions are used as inputs to the downstream task (class prediction), making the pipeline interpretable and less expressive, thus less prone to information leakage.

When compared with soft CBMs and Soft CIBMs:

- Representation:
    - Hard CBMs: Use discrete hard values for concepts (e.g., 0 or 1 for binary concepts).
    - Soft CBMs: Use continuous values (e.g., logits or probabilities).
    - Soft CIBMs: Similar to soft CBMs but use IB to minimize irrelevant information, reducing concept leakage.
- Information Flow:
    - Hard CBMs: Compress information into discrete concept values, which prevents information leakage but risks losing useful details for downstream tasks.
    - Soft CBMs: Retain richer information but are more prone to concept leakage.
    - Soft CIBMs: Balance retaining relevant information while mitigating leakage through the IB framework.
- Interventions:
    - Hard CBMs: Explicitly rely on discrete corrections during interventions, which can have a significant impact.
    - Soft CBMs and CIBMs: Treat interventions as updates to probabilities or logits, which is more expressive, but could induce noise in concepts.

Due to their rigidity, without enough interventions, hard CBMs cannot recover from errors or noise in the predicted concepts because the discrete pipeline does not allow for soft adjustments.

But, as more concepts are corrected, the discrete nature of hard CBMs becomes an advantage together with its independent training: ground truth, hard values fully override noisy predictions, ensuring perfect input for the downstream classifier, which was previously trained also on ground truth concepts from train set.

Soft CBMs and CIBMs, while retaining more information, still rely on probabilistic updates during interventions, which may not fully override noisy concept predictions.

Overall, CIBMs are superior because they combine the advantages of soft representations (expressiveness, better performance) with mechanisms to mitigate concept leakage (robustness, interpretability). Hard CBMs, while conceptually cleaner in avoiding leakage, fail to achieve the same level of downstream performance and adaptability, particularly in more realistic or challenging scenarios.

