# OpenReview forum: "Concepts' Information Bottleneck Models"
_NeurIPS.cc/2025/Conference — Submitted to NeurIPS 2025_

### Official Review · Reviewer_qi7a · 2025-06-03

**Clarity:** 2
**Significance:** 2
**Originality:** 3
**Rating:** 4
**Confidence:** 3

**Summary:**

The paper proposes a Concept Information Bottleneck model, which addresses the information leakage issue in Concept Bottleneck Models while preserving concept and label prediction accuracy through Information Bottleneck regularization.

**Questions:**

If the authors can address the discrepancies in the reported results (Weakness 2) and provide more comprehensive and interpretable evidence of the improvements brought by the CIBM (Weakness 3), I would consider raising my overall evaluation of the paper.

**Ethical Concerns:**

["NO or VERY MINOR ethics concerns only"]

**Final Justification:**

While I cannot fully confirm the conclusions you presented, I also do not have sufficient evidence to suggest that there are serious issues with your experiments. This may require readers to reproduce your code and engage in further discussion with the authors of ProbCBM. Therefore, under the current circumstances, I have decided to raise my score to 4.

**Limitations:**

The rationale for why leakage can be mitigated through IB regularization appears to be heuristic, lacking a formal theoretical justification. However, I do not consider such a theoretical proof to be strictly necessary for the contribution of this paper.

**Quality:**

3

**Strengths And Weaknesses:**

Strengths:

1. The paper is overall well-written and well-organized.

2. The experiments conducted in this paper are comprehensive.

3. The proposed loss function is plug-and-play, making it easily applicable to various CBM models while effectively addressing their leakage issues.

Weaknesses:

1. The performance gains brought by the IB regularization appear to be marginal.

2. Some reported results are inconsistent with prior work. For example, the label accuracy of CBM (SJ) reported in the original CBM paper (ICML 2020, https://arxiv.org/pdf/2007.04612) is around 80% (see Table 1 and Fig. 2), whereas this paper reports a significantly lower value of 70.8%.

3. The authors should provide more detailed evidence to demonstrate how the proposed methods enhance the interpretability of CBMs. The numerical results in Table 2 are insufficient, as it is unclear what the reduction in OIS from 4 to 3 actually implies. For instance, the experiments in https://arxiv.org/pdf/2106.13314 offer a more intuitive demonstration of concept leakage in CBMs. It would be helpful to see a direct comparison between CBM and the proposed CIBM under such experimental setups.

---

> ### Author Rebuttal · Authors · 2025-07-30
>
> > The performance gains brought by the IB regularization appear to be marginal.
>
> The results in concept prediction appear marginal due to the saturation of the baselines. However, **we observe an improvement in class prediction with our proposed IB regularizer compared to the base models**. Additionally, the test-time interventions demonstrate clear advantages of the regularized methods over their base counterparts. Furthermore, our information-plane dynamics analysis indicates that our regularization enhances the information content of the representations.
>
> Thus, **when considering the overall set of results rather than individual ones, the significance of our findings becomes evident**.
>
> ---
> > Some reported results are inconsistent with prior work. For example, the label accuracy of CBM (SJ) reported in the original CBM paper (ICML 2020, https://arxiv.org/pdf/2007.04612) is around 80% (see Table 1 and Fig. 2), whereas this paper reports a significantly lower value of 70.8%.
>
>
> We utilized the ProbCBM official repository to run CBM and ProbCBM, incorporating our regularizers on top of them. Therefore, the difference in CBM results stems from this specific implementation. We **chose this implementation to ensure a fair comparison**, as it maintains the same base architecture, hyperparameters, and learning framework across CBMs. Additionally, we opted to use the implementation of a more advanced CBM, like ProbCBM, rather than starting with a simpler one and constructing the more complex versions on top of it.
>
> As illustrated in Table 1, recent additions to CBMs, such as AR-CBM, CEM, and ProbCBM, yield stronger results compared to the original CBM. Hence, we did not consider this performance difference a major issue, given the fairness of the setup.
>
> Furthermore, we highlight that **inconsistencies exist in reported models across papers**, as some utilize different versions of CBM models without clearly stating which version is used or detailing changes in the architecture. Therefore, **we decided to create our own baselines to ensure fair and comparable models**.
>
> ---
> > The authors should provide more detailed evidence to demonstrate how the proposed methods enhance the interpretability of CBMs. The numerical results in Table 2 are insufficient, as it is unclear what the reduction in OIS from 4 to 3 actually implies. For instance, the experiments in https://arxiv.org/pdf/2106.13314 offer a more intuitive demonstration of concept leakage in CBMs. It would be helpful to see a direct comparison between CBM and the proposed CIBM under such experimental setups.
>
> Our experiments are multi-faceted and intuitive, as evidenced by our results of the reduction in concept leakage. Firstly, **we employ state-of-the-art quantitative metrics to measure and compare concept leakage** (Section 4.2). Our results show that models regularized with the Information Bottleneck (IB) framework exhibit a significant reduction in concept leakage compared to their original counterparts. Next, **we conducted intervention experiments** (Section 4.3), which revealed that correcting concept groups at test time resulted in CIBMs displaying significantly smoother, monotonic increases in accuracy. As well as the **concept quality within the models** (Section 4.4). Additionally, **we analyzed information plane dynamics** (Section 4.5), illustrating how CIBMs maintain higher mutual information with labels (good information) and lower mutual information with raw inputs (irrelevant information), aligning precisely with our theoretical goals.
>
> This **multi-faceted strategy** (combining rigorous quantitative metrics, practical intervention experiments, and theoretical information analyses) **is thoughtfully designed to provide comprehensive and intuitive evidence of leakage reduction from all critical perspectives**.
>
> Due to the limited time available during the rebuttal, we were unable to perform the requested experiments from "Promises and Pitfalls of Black-Box Concept Learning Models." However, while these experiments may enhance the existing evidence, we believe they are not crucial given the ample evidence we have already provided. Nonetheless, we will conduct these experiments and include the results in the final version of the manuscript.
>
> ---
> > The rationale for why leakage can be mitigated through IB regularization appears to be heuristic, lacking a formal theoretical justification. However, I do not consider such a theoretical proof to be strictly necessary for the contribution of this paper.
>
> Our approach begins with a **principled problem that compresses information in the latents concerning the data while maintaining expressiveness regarding the labels**. We believe this establishes a meaningful **connection between leakage and IB regularization through compression**, which is stronger than a mere heuristic. However, we acknowledge that this link is not fully formalized.
>
> While we agree that a comprehensive theoretical proof is valuable, we believe our empirical results and practical implications significantly contribute to the field. Our experiments show substantial improvements in reducing concept leakage, supporting the validity of our approach. We welcome further discussion and recognize the importance of formalizing these connections in future work.

---

> ### Comment · Reviewer_qi7a · 2025-08-01
>
> Thank you for the clarification. However, I still feel confused about the results reported in the paper:
>
>
> **1. CBM Accuracy Too Low**
>
> I have personally reproduced the results of CBM (Joint-CBM on CUB) following the original ICML 2020 paper, and achieving 78–80% label accuracy is quite straightforward. The reported 70.8% accuracy for CBM is much lower than prior work (78–80%). Even with different hyperparameters, it’s easy to reach above 75%. This result is hard to accept as a valid baseline.
>
> **2. Lack of Detailed Explanation**
>
> You mention using a different implementation, but do not specify what changed—backbone, loss weights, training setup, etc. Without details, the cause of the performance drop remains unclear.

---

> > ### Author Response · Authors · 2025-08-01
> >
> > We acknowledge the reviewer's observation that our baseline CBM accuracy on CUB (\~70%) is lower than reported in the original CBM paper (\~78-80%). This discrepancy is due to our deliberate choice of starting point for implementation. Instead of reproducing the original CBM implementation, we adopted the more recent ProbCBM implementation, as described in Kim et al., ICML 2023. The concept and class accuracies reported by ProbCBM authors match our results (see Table 1 of their paper, CBM class accuracy ~70%), confirming the correctness of our implementation. The **main difference between the ProbCBM and original CBM implementations is the number of training epochs**: our setup trains for **100 epochs**, whereas the original method trains for up to 1000 epochs. To the best of our knowledge, all other hyperparameters remain consistent, except for library version differences due to the time gap between implementations.
> >
> > We selected ProbCBM's implementation because it is open-source and recently published, ensuring our results are reproducible and comparable by the broader community. This choice also provides a fair and contemporary comparison by mitigating potential implementation-specific biases. Our baseline numbers closely align with ProbCBM's reported values, reinforcing the accuracy and reliability of our baselines.
> >
> > Finally, please note that **our core contributions are methodologically focused-specifically on introducing and validating the IB regularization**. Given that we use exactly the same baseline (CBM without IB) and apply the IB regularization consistently, **our reported improvements unequivocally demonstrate the relative efficacy of our proposed regularization technique**. The primary focus of our study remains robustly validated: **the IB-regularized models (CIBMs) consistently improve upon their corresponding CBM baselines when all other variables (dataset, code, implementation) remain constant**. Moreover, the same improvement can be seen in the other methods we used.
> >
> > We appreciate the reviewer highlighting this point and will explicitly document this rationale in the final paper version. If the reviewer desires, we can conduct the CBM experiment with a longer training duration and hope to provide results in a few days. Considering our infrastructure constraints, we are not sure how many runs we will be able to provide during the discussion period.

---

> ### Comment · Reviewer_qi7a · 2025-08-01
>
> I remain confused by your response, so I re-checked both the original CBM and ProbCBM papers. The original CBM paper (ICML 2020) uses **InceptionV3** as the backbone (“by fine-tuning an Inception-v3 network”), while ProbCBM uses **ResNet18** (“ResNet18 is used as a backbone in all models”). I believe the difference in backbones is the main reason for the discrepancy between the two papers, which is also consistent with my own experimental experience. However, in your paper, **you clearly state that InceptionV3 is used for CUB** (Line 535). This implies your backbone should be aligned with the original CBM paper, and thus should not result in such a large performance drop.
>
> I don’t think I’m questioning the importance of your work, but at this point, I’m more concerned about whether the reported results are solid.

---

> > ### Author Response · Authors · 2025-08-02
> > **The main reason for the difference with the original CBM is the number of epochs used for training**
> >
> > We sincerely thank the reviewer for their careful and thorough evaluation of our paper, and we acknowledge the validity and importance of the issue they've raised.
> >
> > After carefully reviewing our experimental details, we confirm the reviewer's observation: although we used the InceptionV3 backbone (consistent with the original CBM paper), **our training regimen followed ProbCBM’s shorter 100-epoch schedule rather than the original CBM paper’s 1000-epoch schedule**. This discrepancy in training epochs accounts for the difference in our reported baseline accuracy (\~70%) compared to the original CBM accuracy (\~78–80%).
> >
> > To address this issue, we have trained a CBM SJ InceptionV3 model using the original 1000-epoch regime and compared it to the values reported in our paper, as shown in the table below. The discrepancy observed by the reviewer is primarily due to the reduced number of training epochs. **Models trained for 1000 epochs are closer to the original reported values in CBM.** Additionally, the table demonstrates that **our regularizers still improve upon the baseline**.
> >
> > | Method | Epochs | Concept | Class |
> > | - | - | - | - |
> > | CBM | 100 | 0.956 | 0.708 |
> > | CBM + IB-B | 100 | 0.958 | 0.725 |
> > | CBM + IB-E | 100 | 0.959 | 0.729 |
> > | **CBM** | 1000 | 0.957 | 0.795 |
> > | **CBM + IB-B** | 1000 | 0.957 | 0.820 |
> > | **CBM + IB-E** | 1000 | 0.957 | 0.822 |
> >
> > Additionally, we trained a CBM SJ model using the ResNet18 backbone under the ProbCBM 100-epoch training regime, and we report the results in the table below alongside our original values. As demonstrated, the difference between the two backbones is similar, supporting our explanation that the main disparity in accuracy is due to the number of training epochs used.
> >
> > | Method | Backbone | Concept | Class |
> > | - | - | - | - |
> > | CBM | Inception | 0.956 | 0.708 |
> > | CBM + IB-B | Inception | 0.958 | 0.725 |
> > | CBM + IB-E | Inception | 0.959 | 0.729 |
> > | CBM | Resnet | 0.951 | 0.683 |
> > | CBM + IB-B | Resnet | 0.959 | 0.714 |
> > | CBM + IB-E | Resnet | 0.958 | 0.702 |
> >
> > Thus, we reiterate that our core methodological contribution (the effectiveness of IB regularization in reducing concept leakage) remains robust and valid. Additionally, **we reaffirm that our baselines are valid and solid**, with the primary difference being the number of training epochs w.r.t. the original CBM paper. We will clarify these accuracy differences in the final version of the paper.

---

> ### Comment · Reviewer_qi7a · 2025-08-03
> **I believe models need to be sufficiently trained.**
>
> Thank you for your response. However, **I still find it difficult to believe that your current experimental results are valid and solid.** Based on our previous exchanges, it has become clear that you trained the InceptionV3 backbone for only 100 epochs, which demonstrably leads to **insufficient training**. I believe models need to be sufficiently trained. Since there's a 10-point difference between 100 and 1000 epochs, you shouldn't just report the results from 100 epochs. ProbCBM uses ResNet18, and I suspect it can converge within 100 epochs, which is why the authors in ProbCBM only trained for 100 epochs. However, you're currently using InceptionV3, and it's been demonstrated that models trained for 100 epochs still have significant room for improvement.
>
> Furthermore, your experimental setup **neither aligns with the original CBM paper nor with ProbCBM as you initially claimed.** **Your statement in the first rebuttal, "we opted to use the implementation of a more advanced CBM, like ProbCBM," was incorrect**, as you used a different backbone (InceptionV3) than ProbCBM (ResNet18). While you've provided new experiments for CBM, it raises a crucial question: did you also use InceptionV3 as the backbone and train for only 100 epochs for the other models reported on the CUB dataset?
>
> Given our discussions, it appears that the authors may not have a thorough understanding of their own experimental settings, nor how they relate to prior work, when the paper was initially written. I respectfully believe it's imperative that the authors **comprehensively re-examine all their experimental settings** to ensure the reliability and consistency of their reported results.
>
> Therefore, please forgive me that I cannot recommend this paper for publication at NeurIPS at this time.

---

> > ### Author Response · Authors · 2025-08-03
> >
> > We sincerely appreciate the reviewer's meticulous scrutiny and apologize for any confusion in the setup description in this rebuttal. However, we respectfully disagree with the conclusion that our experiments are not valid.
> >
> > The additional experiments conducted (using InceptionV3 trained for 1000 epochs and ResNet18 trained for 100 epochs) have confirmed that our IB-based regularization consistently enhances performance compared to corresponding CBM baselines. **These experiments demonstrated that the discrepancy in accuracy is due to the number of training epochs used rather than an underlying methodological issue.** We emphasize that our regularizers still show improvement even at 1000 epochs. Training for fewer epochs does not imply insufficient training; it simply presents results at different stages of training. Critically, our IB-regularized models consistently outperform baselines at each corresponding stage. While longer training with more epochs may offer extended insights, it is not necessary.
> >
> > Moreover, we underscore that more recent models such as CEM, ProbCBM and AR-CBM exhibit similar improvements and as we have shown the improvements through our regularizers were consistently observed across all of them, clearly demonstrating the broad robustness and general applicability of our method.
> >
> > We highlight that the confusion with the CBM setup is an exception (given its correct documentation in the paper), and all other setups are accurately documented as well.
> >
> > Thus, while we fully acknowledge the reviewer's valuable feedback, we maintain that our methodological contribution—the effectiveness of IB-based regularization in reducing concept leakage and improving performance—is robust, clearly demonstrated, and validated across multiple experimental settings.

---

> > > ### Comment · Reviewer_qi7a · 2025-08-03
> > >
> > > When I re-checked the ProbCBM paper, I found that the experimental results you reported for CBM, ProbCBM, and CEM are **completely identical** to those in the original ProbCBM paper, both in terms of label and concept accuracy and std. So I have reason to believe that **these baseline results were copied from the ProbCBM paper.**
> > >
> > > However, the problem is that **ProbCBM uses ResNet18 as the backbone**, while **your CBM-IB, ProbCBM-IB, and CEM-IB are reported to use InceptionV3**. I think such a comparison is completely unreasonable and does not convince me that your experiments are solid. You need to re-check and re-conduct all experiments that are affected.
> > >
> > > I believe this issue is not even about insufficient training, but rather that during your experiments, **the backbones of X and X-IB were not aligned**, where X represents the baseline model.

---

> > > > ### Author Response · Authors · 2025-08-04
> > > >
> > > > We appreciate the reviewer's follow-up and closer evaluation of our results.
> > > >
> > > > To clarify: (a) we have reproduced all results, and (b) in each comparison between the baseline models (X vs. X-IB), X remains unchanged, including the backbone, training epochs, and all other hyperparameters.
> > > >
> > > > Additionally, our results presented in this rebuttal demonstrate that IB regularization continues to improve the methods even with longer training epochs.
> > > >
> > > > We apologize for any confusion caused by the wording regarding the backbones used in the rebuttal.

---

> ### Comment · Reviewer_qi7a · 2025-08-04
>
> ## Why are your results using InceptionV3 completely identical to those in the original ProbCBM paper using ResNet18, including CBM, ProbCBM, and CEM, across both concept and label accuracy and standard deviation?
>
> | Method    | Concept Accuracy (mean ± std) | Label Accuracy (mean ± std) |
> |-----------|-------------------------------|------------------------------|
> | CBM       | 0.956 ± 0.001                 | 0.708 ± 0.006                |
> | CEM       | 0.954 ± 0.001                 | 0.759 ± 0.002                |
> | ProbCBM   | 0.956 ± 0.001                 | 0.718 ± 0.005                |
>
> The results in Table 1 of the ProbCBM paper (https://arxiv.org/pdf/2306.01574) are identical to the ones in Table 1 of your paper. But different backbones are used.

---

> > ### Author Response · Authors · 2025-08-04
> >
> > We believe there has been a miscommunication regarding the backbones used in our experiments. To clarify, here are the specific details:
> >
> > - For CBMs, the original paper used Inception, and we did the same, although our code is sourced from the ProbCBM paper.
> > - ProbCBM employed ResNet18, and we utilized their configuration and code for our experiments.
> > - CEM used ResNet34 in their paper, which we also adopted, using code from the CEM repository.
> > - AR-CBM used Inception, which we followed, using code from the AR-CBM repository.
> >
> > Our results for ProbCBM, CEM, and AR-CBM were similar to the values reported in the ProbCBM paper, falling within a standard deviation. We reproduced the original paper's values for completion, as it is standard practice to report previous results given the same experimental setup. However, we did not follow the same practice for the CBMs due to changes in the number of training epochs. We did not view this as an issue since **our primary focus was explicitly on comparing each baseline model directly against its corresponding IB-regularized variant** to assess the effectiveness of IB regularization.
> >
> > To fully address this discrepancy, **we conducted and reported additional experiments (in a previous rebuttal comment)**: (a) We trained CBM and our IB-regularized CBMs for the original 1000 epochs using InceptionV3, achieving accuracy close to the original CBM paper (~78-80%). We also repeated our CBM experiments using the ProbCBM ResNet18 backbone setup for 100 epochs, which resulted in slightly different yet similar values.
> >
> > While our experimental setups varied across methods, each comparison between a baseline model and its IB-regularized variant was entirely consistent internally, which is crucial for validating the effectiveness of our IB regularization approach.

---

> > > ### Comment · Reviewer_qi7a · 2025-08-05
> > >
> > > ## Your results using different backbones are **completely identical** to those reported in the original ProbCBM paper with ResNet18—this includes CBM, ProbCBM, and CEM, across both concept accuracy, label accuracy, and standard deviation.
> > >
> > > Please confirm whether this is indeed the case. Personally, I find this result difficult to believe, but I will respect the AC's final decision.

---

> > > > ### Author Response · Authors · 2025-08-05
> > > >
> > > > We confirm that the backbones used in each method corresponds to the original ones and are the same used in the regularized versions.  Our results for ProbCBM, CEM, and AR-CBM were similar to the values reported in the ProbCBM paper, falling within a standard deviation. We reproduced the original paper's values for completion, as it is standard practice to report previous results given the same experimental setup. However, we did not follow the same practice for the CBMs due to changes in the number of training epochs. We did not view this as an issue since our primary focus was explicitly on comparing each baseline model directly against its corresponding IB-regularized variant to assess the effectiveness of IB regularization.

---

> > > > > ### Comment · Reviewer_qi7a · 2025-08-09
> > > > >
> > > > > Thank you for your response. While I cannot fully confirm the conclusions you presented, I also do not have sufficient evidence to suggest that there are serious issues with your experiments. **This may require readers to reproduce your code and engage in further discussion with the authors of ProbCBM.** Therefore, under the current circumstances, I have decided to raise my score to 4.

---

### Official Review · Reviewer_f3CC · 2025-07-02

**Clarity:** 3
**Significance:** 2
**Originality:** 3
**Rating:** 3
**Confidence:** 4

**Summary:**

This paper proposes CIBM, an extension of Concept Bottleneck Models (CBMs) that incorporates an informational regularization term that is borrowed from the Information Bottleneck (IB) method of Tishby et al. (1999). The paper evaluates several variants of this method and shows that the BI-regularization term yields improved performance in terms of test time accuracy and concept leakage.

**Questions:**

My main concern is that it's not entirely clear what is the big-picture contribution and/or deep new insight that this method can offer, beyond the technical results that seem somewhat incremental. If the authors can clarify this matter, that would be extremely helpful for assessing whether this paper merits publication at a top-tier ML venue like NeurIPS.

**Ethical Concerns:**

["NO or VERY MINOR ethics concerns only"]

**Final Justification:**

While this is an interesting submission, I tend to think it's contribution is too incremental for a venue like NeurIPS.

**Limitations:**

yes

**Quality:**

2

**Strengths And Weaknesses:**

**Strengths**
- The paper proposes an interesting new method, with a substantial set of experiments. Overall, I'm inclined to think that this method has merit and could potentially make a valuable contribution to the NeruIPS community (but see my main question below).
- This work is original in the sense that it combined existing techniques in a novel way and the motivation for integrating IB with CBMs is clearly conveyed. I find the concept leakage analysis particularly useful and directly related to the IB motivation.
- The paper is well organized and clearly written.

**Weaknesses**
- While the paper presents a substantial set of experiments, many of them are not clearly motivated or insightful. For example, the paper considers 12 variants -- two bounds and 6 training settings (HJ / HI / SJ / Prob / CEM / AR ) -- but it's unclear why they are all equally interesting or useful. The current presentation of the paper lacks proper motivation of all these variants, giving an impression they are all considered simply because it's possible, not necessarily because they give unique insight into the problem/solution.
- Relatedly, the E-CIB bound assumes that the concepts entropy $H(C)$ is constant, while this entropy is not constant given the implementation details in the appendix. Thus, this bound, even though it performs better than the SUB-CIB bound, does not seem to be well motivated, or at least this issue is not addressed in the main text.
- The choice of $\beta$ values seems quite arbitrary as well. Why these two particular values? The paper seems to incorrectly treat $\beta$ as a constant rather than a free (hyper)parameter.
- The main text is not self-contained and relies too much on the appendix for key details and results. For example, the dataset are not explained at all in the main text, the main results for sections 4.5 and 4.6 are only in the appendix.

**Minor points and suggestions**
- I think the main text would benefit from a concrete example to motivate and illustrate the method.
- Table 1 is pretty hard to digest, I would consider visualizing these results with bar plots and then maybe moving the detailed table to the appendix.
- Typos:
   - lines 56-57: use either $K$ or $k$ in both instances
   - line 125: shouldn't this be $I(X;Z)$? as in Eq.1?
   - Eq 3: missing $E_{p(c)}$ for the third term

---

> ### Author Rebuttal · Authors · 2025-07-30
>
> > My main concern is that it's not entirely clear what is the big-picture contribution and/or deep new insight that this method can offer, beyond the technical results that seem somewhat incremental. If the authors can clarify this matter, that would be extremely helpful for assessing whether this paper merits publication at a top-tier ML venue like NeurIPS.
>
> The significance of our CIBM framework goes beyond incremental technical contributions and represents a fundamental conceptual advance (as we detail below).
>
> Traditional CBMs introduced intermediate concepts primarily as interpretable components; however, they lacked any rigorous mechanism to guarantee that these concepts would contain only meaningful, minimal, and task-relevant information. In contrast, our work **explicitly integrates the IB principle directly into the concept layer**. This represents a substantial, principled, theoretical reframing: **moving interpretability from a heuristic add-on toward a structurally enforced, theoretically grounded property of the model itself**.
>
> From the perspective of information leakage, previous research primarily viewed leakage as an empirical inconvenience or a purely technical issue to be solved by modifying architectures or concept definitions. Our work offers a clear conceptual shift: **we recognize and address leakage explicitly as a fundamental information-theoretic issue**. Specifically, leakage arises because traditional CBMs allow irrelevant information from inputs to be encoded unintentionally into the concept layer. Instead of incremental architectural tweaks, **we demonstrate that explicitly constraining mutual information between inputs and concepts through IB regularization solves this leakage problem at its root**. This approach yields better concept representations, substantially improving their reliability for real-world interpretability and intervention tasks, as our empirical results convincingly show.
>
> Moreover, the **IB-based approach we introduce is broadly generalizable and not limited to a single CBM architecture or variant**. Our empirical evaluations demonstrate consistent improvements across numerous concept-based frameworks, including CEM, ProbCBM, and AR-CBM. Therefore, our approach does not merely offer incremental improvement on a single model class; instead, it provides a robust, foundational methodology that can be adopted broadly across the interpretability community.
>
> ---
> > While the paper presents a substantial set of experiments, many of them are not clearly motivated or insightful. For example, the paper considers 12 variants -- two bounds and 6 training settings (HJ / HI / SJ / Prob / CEM / AR ) -- but it's unclear why they are all equally interesting or useful. The current presentation of the paper lacks proper motivation of all these variants, giving an impression they are all considered simply because it's possible, not necessarily because they give unique insight into the problem/solution.
>
> While the paper presents a substantial set of experiments, we would like to clarify that **each of these methods serves as contemporary and meaningful baselines to contextualize our contributions**. The inclusion of the variants was essential to evaluate our method against commonly used foundational CBMs and methods specifically designed to reduce information leakage.
>
> By conducting a comprehensive comparison, we aimed to establish clear benchmarks and quantify the degree of improvement our method offers. We included traditional CBMs (CBM-HJ, CBM-HI, CBM-SJ) as foundational baselines to demonstrate effectiveness, even though these models were not explicitly designed to address leakage. In addition, we compared our method to Auto-Regressive CBMs (AR-CBM), which specifically aim to minimize information leakage, making this a critical comparison for directly showcasing our approach's effectiveness. Moreover, we considered Probabilistic CBMs (ProbCBM) and Concept Embedding Models (CEM), which address leakage indirectly through enhanced concept representations, quality, or uncertainty modeling. These comparisons provide a thorough evaluation against state-of-the-art methods and demonstrate the generalization capabilities of our proposal.
>
> In summary, **our choice of variants was driven by the need to present meaningful insights rather than merely the possibility of including them**.
>
> ---
> > Relatedly, the E-CIB bound assumes that the concepts entropy H(C) is constant, while this entropy is not constant given the implementation details in the appendix. Thus, this bound, even though it performs better than the SUB-CIB bound, does not seem to be well motivated, or at least this issue is not addressed in the main text.
>
> In deriving the E-CIB bound, we simplified the formulation by treating the entropy of the concepts as approximately constant. On one hand, **we assume that the concepts entropy was constant since it depends on the prior distribution of the concepts**. On the other hand, **this simplification is also empirically supported** by our results (see below), which confirm that concept entropy indeed rapidly stabilizes early in training, exhibiting minimal fluctuation thereafter.
>
> We computed the concepts entropy over the epochs and see that the entropy quickly stabilizes (after epoch 3) and doesn't significantly change, as shown below:
> | Epoch | 0 | 1 | 2 | 3 | 4 | 5 | ... |
> | --- | --- | --- | --- | --- | --- |  --- |  --- |
> | H( C ) | 13.51 | 4.62 | 2.09 | 2.08 | 2.08 | 2.08 | 2.08 |
>
> In the final version, we will add a plot showing the curve and more epochs.  However, due to the text restrictions of this rebuttal, we cannot show the figure fully.
>
> ---
> > The choice of $\beta$ values seems quite arbitrary as well. Why these two particular values? The paper seems to incorrectly treat $\beta$ as a constant rather than a free (hyper)parameter.
>
> As noted by the reviewer, $\beta$ is a hyperparameter. In our paper, we reported two values derived from initial experiments. Below, we provide additional values illustrating the behavior of CBM SJ + IB_E. As shown, $\beta = 0.5$ achieves the highest class accuracy, which supports our choice. While one might argue for exploring different values of $\beta$, this approach could lead to unfair comparisons between experiments and increase costs. Therefore, we decided to use a single value for all experiments. We will report these extended experiments in the final version of the paper.
>
> | $\beta$ | Concept | Class |
> | ------- | ------- | ----- |
> | 0.10 | 0.958 | 0.728 |
> | 0.20 | 0.958 | 0.727 |
> | 0.25 | 0.958 | 0.727 |
> | 0.50 | 0.958 | 0.729 |
> | 0.75 | 0.958 | 0.724 |
> | 0.90 | 0.958 | 0.726 |
>
> ---
> > The main text is not self-contained and relies too much on the appendix for key details and results. For example, the dataset are not explained at all in the main text, the main results for sections 4.5 and 4.6 are only in the appendix.
>
> Due to space constraints in the paper, we had to move details into the appendix. While we agree with the reviewer that these details will be better in the main paper, we compromised by having the main discussion in the paper while having additional details in the appendix.
>
>
> ---
> > I think the main text would benefit from a concrete example to motivate and illustrate the method.
>
> We thank the reviewer for the suggestion.  We will review the final version to add an example that illustrates the method that is tied to Fig. 1.
>
> ---
> > Table 1 is pretty hard to digest, I would consider visualizing these results with bar plots and then maybe moving the detailed table to the appendix.
>
> We thank the reviewer for the suggestion.  We will consider it for the final version of the paper.
>
> ---
> > Typos: lines 56-57: use either K or k in both instances; Eq 3: missing E_p(c) for the third term
>
> We thank the reviewer for the detailed comments.  We will fixed them in the final version of the paper.
>
> ---
> > Typos: line 125: shouldn't this be I(X; Z) as in Eq.1?
>
> In this line, we refer to our final objective in Eq. 2.  But we can see the confusion since this could refer to the original objective (1) as well.  We will review our writing to make this clear in the final version of the paper.

---

> > ### Comment · Reviewer_f3CC · 2025-08-04
> >
> > I thank the authors for their detailed response to my comments. While I appreciate all the hard work that went into this submission, I still feel it's contribution is somewhat limited and the author's rebuttal didn't present any new evident that convinced me otherwise. To clarify, the IB formulation is indeed elegant and insightful, however it's not new or unique to this work. For this work to be insightful, I was looking for some interesting observations as to the nature of the solution compared to traditional CBMs, beyond marginal quantitative improvements. In the IB framework, such insight typically arises from varying $\beta$, but the paper doesn't explore this and the author's rebuttal suggests that this key feature of IB might not be particularly useful in this context, which makes this work rather incremental in my view.

---

> ### Author Response · Authors · 2025-08-04
>
> Thank you for your feedback on the rebuttal. However, we respectfully disagree with the notion that applying the IB framework in a new setting lacks value.
>
> > [...] however it's not new or unique to this work.
>
> We acknowledge that the IB framework itself is not novel; this is neither our contribution nor our claim.
>
> > For this work to be insightful, I was looking for some interesting observations as to the nature of the solution compared to traditional CBMs, beyond marginal quantitative improvements.
>
> We agree that exploring interesting insights from our framework compared to traditional CBMs would be beneficial for the community. However, our work represents a first step in this direction. To our knowledge, there are no other studies investigating the use of IB in CBMs, making **the exploration of IB regularization and its effects on current models noteworthy**.
>
> > In the IB framework, such insight typically arises from varying $\beta$ [...]
>
> We agree that varying $\beta$ often yields insights, and we demonstrated that the changes are minimal in this case. Furthermore, we provided the information plane dynamics to illustrate the differences in training between methods. As mentioned, this is ongoing work that extends beyond the scope of our current proposal. Our study serves as a foundational effort for future research, further showcasing its significance and relevance.

---

### Official Review · Reviewer_Pg29 · 2025-07-03

**Clarity:** 4
**Significance:** 3
**Originality:** 4
**Rating:** 4
**Confidence:** 4

**Summary:**

This paper addresses the issue of concept leakage in Concept Bottleneck Models (CBMs), which are representative interpretable models. To mitigate this problem, the authors propose applying an Information Bottleneck (IB) regularizer to the concept layer. The proposed method is shown to be effective across various CBM families.

**Questions:**

- Can the authors provide a comparison with previous methods that also aim to reduce information leakage in CBMs or similar models?

## Additional Suggestions:
- In Figure 3, the improved TTI performance due to the IB regularizer is clearly visible for CEM, but less so for other methods. Consider revising the visualization to make this improvement more apparent across models.
- In Figure E.1, the fact that the first and second columns correspond to different datasets is only indicated in the last row, which may be confusing. It would help to clarify this more explicitly.

**Ethical Concerns:**

["NO or VERY MINOR ethics concerns only"]

**Final Justification:**

Some of my initial concerns have been addressed by the authors' rebuttal. However, with regard to concerns about other existing works, the authors only mentioned that they would address them, without providing a clear response. While this issue is not critical enough to justify a recommendation for rejection, it also makes it difficult to justify raising the score.

**Limitations:**

Yes

**Paper Formatting Concerns:**

No concern

**Quality:**

3

**Strengths And Weaknesses:**

## Strengths:
- The paper clearly identifies the problem and proposes a well-motivated solution.
- It explores multiple design choices and supports them with theoretical justifications.
- The proposed method improves both prediction performance and mitigates concept leakage, as demonstrated through comprehensive experiments.
- The authors also identify limitations in existing evaluation metrics and propose a new metric that better assesses the quality of the concept set, analyzing the results thoroughly.

## Weaknesses:
- While the method demonstrates effectiveness when applied to basic CBM families, there is no experimental comparison with CBMs that use other forms of information bottleneck techniques. Although the differences in approach are discussed conceptually, empirical comparisons would strengthen the evaluation of information leakage mitigation.

---

> ### Author Rebuttal · Authors · 2025-07-30
>
> > While the method demonstrates effectiveness when applied to basic CBM families, there is no experimental comparison with CBMs that use other forms of information bottleneck techniques. Although the differences in approach are discussed conceptually, empirical comparisons would strengthen the evaluation of information leakage mitigation.
>
> To the best of our knowledge, **at the submission of this work, there was no evidence explicitly integrating an Information Bottleneck loss into a concept bottleneck model's training objective**. A recent search revealed the pre-print titled "There Was Never a Bottleneck in Concept Bottleneck Models," uploaded in June 2025, indicating that it is concurrent work according to NeurIPS regulations. Furthermore, this pre-print does not conduct traditional experiments that would enable direct comparison.
>
> ---
> > Can the authors provide a comparison with previous methods that also aim to reduce information leakage in CBMs or similar models?
>
> We would like to emphasize that **we already conducted a comprehensive comparison, including methods that aim to reduce leakage, both directly and indirectly**. Specifically, we compared our method against traditional CBMs (CBM-HJ, CBM-SJ, CBM-HI) as foundational baselines to establish clear benchmarks and quantify the improvement our method offers, despite these models not being explicitly designed to address leakage. Additionally, our method was compared to **Auto-Regressive CBMs (AR-CBM)**, which **specifically target minimizing information leakage**, directly demonstrating the effectiveness of our approach. Furthermore, we considered Probabilistic CBMs (**ProbCBM**) and Concept Embedding Models (**CEM**), which **address leakage indirectly** through improved concept representations, quality, or uncertainty modeling.
>
> These serve as contemporary and meaningful baselines to further contextualize our contributions. Therefore, **while additional experiments involving other methods that aim to reduce information leakage would be valuable, they are not strictly necessary**. Our current experiments effectively demonstrate the information leakage associated with our proposal relative to other methods that also strive to mitigate it.
>
> ---
> > In Figure 3, the improved TTI performance due to the IB regularizer is clearly visible for CEM, but less so for other methods. Consider revising the visualization to make this improvement more apparent across models.
>
> We thank the reviewer for the suggestion.  We will review the visualization to follow the advice.
>
> ---
> > In Figure E.1, the fact that the first and second columns correspond to different datasets is only indicated in the last row, which may be confusing. It would help to clarify this more explicitly.
>
> We thank the reviewer for spotting this mistake.  We will fix this issue in the final version of the paper.

---

> ### Comment · Reviewer_Pg29 · 2025-08-04
>
> Thank the authors for addressing my concerns.
>
> While I acknowledge that the models primarily compared in the paper—CBM-HJ, CBM-SJ, and CBM-HI—are indeed among the most representative approaches proposed to reduce information leakage in concept bottleneck models, there are other relevant works that have also addressed this issue as follows:
>
> [1] Sun, Ao, et al. "Eliminating information leakage in hard concept bottleneck models with supervised, hierarchical concept learning." arXiv preprint arXiv:2402.05945 (2024).
>
> [2] Parisini, Enrico, et al. "Leakage and interpretability in concept-based models." arXiv preprint arXiv:2504.14094 (2025).
>
> Even if [2] does not introduce an explicit mitigation strategy, but it clearly highlights the direction and importance of reducing leakage in concept-based models.
>
> In addition, although the authors claim to have performed a comprehensive comparison with CBM-HJ, CBM-SJ, and CBM-HI, the key evaluation related to the paper’s central claim—i.e., reduction in concept leakage, as shown in Table 2—only includes CBM-SJ (among three) as a baseline. Even if the other methods do not show high performance compared to CBM-SJ, it is important to evaluate all three models side-by-side in this table to convincingly demonstrate the effectiveness of the proposed approach in reducing information leakage (given that CBM-HI was specifically proposed as the most conservative approach to prevent information leakage).

---

> > ### Author Response · Authors · 2025-08-06
> >
> > > [...] there are other relevant works that have also addressed this issue as follows [...]
> >
> > We thank the reviewer for the references.  We will make sure to include these works in the final version of the paper.
> >
> > > [...] it is important to evaluate all three models side-by-side in this table to convincingly demonstrate the effectiveness of the proposed approach in reducing information leakage [...]
> >
> > We have added the requested methods to the table, and we reproduce it below for completeness.
> >
> > | Model             | Complete CS OIS | Complete CS NIS | Selective Drop-out CS OIS | Selective Drop-out CS NIS | Random Drop-out CS OIS | Random Drop-out CS NIS |
> > |-------------------|-----------------|-----------------|---------------------------|---------------------------|------------------------|------------------------|
> > | CBM (SJ)          | 4.69 ± 0.43     | 66.25 ± 2.31    | 16.29                     | 78.39                     | 12.97 ± 0.78           | 74.19 ± 1.04           |
> > | **CBM (SJ) + IBB**| 2.16 ± 0.13     | 61.67 ± 1.92    | 13.09                     | 73.40                     | 10.59 ± 1.48           | 71.38 ± 0.89           |
> > | **CBM (SJ) + IBE**| 2.10 ± 0.17     | 62.36 ± 2.04    | 13.22                     | 72.88                     | 11.00 ± 1.45           | 72.03 ± 1.05           |
> > | CBM (HJ)          | 4.31 ± 0.57     | 65.03 ± 1.84    | 15.71                     | 77.02                     | 13.10 ± 1.20           | 73.09 ± 2.12           |
> > | **CBM (HJ) + IBB**| 2.59 ± 0.87     | 59.84 ± 0.85    | 12.53                     | 71.09                     | 10.36 ± 0.80           | 70.00 ± 1.35           |
> > | **CBM (HJ) + IBE**| 2.30 ± 0.68     | 60.13 ± 2.15    | 12.64                     | 70.89                     | 10.31 ± 0.56           | 70.43 ± 2.10           |
> > | CBM (HI)          | 3.77 ± 0.58     | 65.15 ± 2.03    | 14.27                     | 74.16                     | 12.31 ± 1.04           | 71.78 ± 1.86           |
> > | **CBM (HI) + IBB**| 3.68 ± 0.73     | 64.95 ± 1.60    | 14.04                     | 73.18                     | 12.63 ± 1.37           | 70.92 ± 2.04           |
> > | **CBM (HI) + IBE**| 3.52 ± 0.94     | 63.39 ± 1.28    | 14.05                     | 72.01                     | 12.85 ± 2.18           | 70.17 ± 1.94           |
> > | CEM               | 8.74 ± 0.30     | 75.41 ± 3.83    | 20.85                     | 80.19                     | 18.31 ± 0.09           | 76.56 ± 2.00           |
> > | **CEM + IBB**     | 6.11 ± 0.24     | 70.02 ± 2.21    | 17.22                     | 76.67                     | 14.10 ± 0.42           | 72.68 ± 2.38           |
> > | **CEM + IBE**     | 6.14 ± 0.31     | 71.28 ± 1.90    | 18.00                     | 75.84                     | 14.02 ± 0.37           | 73.88 ± 1.97           |
> > | AR-CBM            | 3.90 ± 0.27     | 62.30 ± 1.52    | 14.16                     | 63.40                     | 12.58 ± 0.86           | 60.86 ± 1.32           |
> > | **AR-CBM + IBB**  | 2.83 ± 0.27     | 59.87 ± 1.52    | 10.97                     | 59.72                     | 10.20 ± 0.55           | 56.28 ± 1.33           |
> > | **AR-CBM + IBE**  | 2.70 ± 0.19     | 60.04 ± 0.98    | 11.05                     | 60.85                     | 11.03 ± 1.02           | 57.21 ± 0.85           |
> > | ProbCBM           | 4.30 ± 0.10     | 64.22 ± 1.04    | 16.01                     | 76.92                     | 13.81 ± 0.21           | 75.01 ± 0.86           |
> > | **ProbCBM + IBB** | 2.53 ± 0.46     | 60.35 ± 2.01    | 13.11                     | 72.86                     | 10.34 ± 0.55           | 70.96 ± 1.91           |
> > | **ProbCBM + IBE** | 2.61 ± 0.53     | 59.86 ± 1.83    | 13.07                     | 73.06                     | 11.20 ± 0.68           | 71.20 ± 2.07           |

---

### Official Review · Reviewer_NahS · 2025-07-06

**Clarity:** 3
**Significance:** 3
**Originality:** 3
**Rating:** 3
**Confidence:** 3

**Summary:**

This paper proposes Concept Information Bottleneck Models (CIBM), a new theoretically inspired architecture to improve concept bottleneck models and reduce concept leakage by directly Incentivizing large mutual information between C and Y, while minimizing mutual information between X and C. With a range of experiments, they show this improves many features of CBMs across different architectures such as Class and concept accuracy and intervention effectiveness.

**Questions:**

- In eq. 5 I don’t see any term corresponding to the I(X; C) term, is this correct? If yes why is this the case?
- Why is the black-box accuracy so high on CUB? This seems higher than ones reported in other papers, for example the original CBM[12] paper reports 82.5% class accuracy. This leaves a large gap between CBM methods and black-box methods.
- Similarly, why are the CBM results reported(SJ) much lower than original CBM paper (70.8% vs 80.1%)?
- How are the results on sequential CBMs? I think the most commonly used CBM type is soft sequential but I don’t see any results on that.
- What are the leakage results (Table 2) of IB_E?
- Is the number for Table 3 AUC IB_B with 8 corrupt correct? Seems not in line with the trends?

**Ethical Concerns:**

["NO or VERY MINOR ethics concerns only"]

**Final Justification:**

While the rebuttal has clarified some of my questions, reading the response and in particular discussion with Reviewer qi7a has significantly reduced my trust in the reported results. In particular it seems clear that at least some of the reported results were using a different backbone than the one mentioned in the paper, and that the baselines were very poorly optimized. If you can improve CBM accuracy by over 10% by training for more epochs then this is clearly not a fully trained model and improving accuracy by 2% over a poorly optimized model is not an interesting finding. I believe the experiments need to be redone with tuned hyperparameters and correct backbones. As a result I am decreasing my score to 3.

**Limitations:**

Yes

**Quality:**

1

**Strengths And Weaknesses:**

Strengths:
- Paper is well motivated and theoretically justified
- Overall clearly written and easy to follow, but a few places could be improved
- Impressive results on concept leakage and intervention effectiveness
- Pretty good results in terms of accuracy improvement


Major Weaknesses:
- Some weirdness in experimental results (see Questions), for example baseline CBM performance is lower than reported on the original CBM paper [12].

Minor Weaknesses:
- Not convinced by the measure of concept-set goodness(4.4), seems like this is closely connected to CBM quality and in the paper it is used to compare different CBMs with the same concept sets, not different concept sets. If the goal is to measure the concept set quality I think the metric should not depend on a particular CBM model.
- Brief introduction in main text would be useful for several things introduced such as the datasets and the Concept Leakage metrics would be helpful.
- The introduction categorization into four groups is confusing, what exactly are the four groups? Also seems like these groupings are not mutually exclusive, one method could belong to multiple of these groups, for example post-hoc explanations can be model-agnostic or model-specific.
- Fig 3 has too many plots in one small fig making it impossible to see all the lines
- Some qualititative examples showing interpretable decisions on different datasets would be helpful

---

> ### Author Rebuttal · Authors · 2025-07-30
>
> > Some weirdness in experimental results (see Questions), for example baseline CBM performance is lower than reported on the original CBM paper [12].
> >
> > Why is the black-box accuracy so high on CUB? This seems higher than ones reported in other papers, for example the original CBM[12] paper reports 82.5% class accuracy. This leaves a large gap between CBM methods and black-box methods.
>
> The black-box accuracy comes from our own execution of the original CBM model [12] minus the concept losses trained supervisedly.  We report this as a black-box baseline since the original CBM paper [12] uses a different backbone and thus it is not fully comparable.
>
> ---
> > Similarly, why are the CBM results reported (SJ) much lower than original CBM paper (70.8% vs 80.1%)?
>
> We utilized the ProbCBM official repository to run CBM and ProbCBM, incorporating our regularizers on top of them. Therefore, the difference in CBM results stems from this specific implementation. We **chose this implementation to ensure a fair comparison**, as it maintains the same base architecture, hyperparameters, and learning framework across CBMs. Additionally, we opted to use the implementation of a more advanced CBM, like ProbCBM, rather than starting with a simpler one and constructing the more complex versions on top of it.
>
> As illustrated in Table 1, recent additions to CBMs, such as AR-CBM, CEM, and ProbCBM, yield stronger results compared to the original CBM. Hence, we did not consider this performance difference a major issue, given the fairness of the setup.
>
> Furthermore, we highlight that inconsistencies exist in reported models across papers, as some utilize different versions of CBM models without clearly stating which version is used or detailing changes in the architecture. Therefore, **we decided to create our own baselines to ensure fair and comparable models**.
>
> ---
> > How are the results on sequential CBMs? I think the most commonly used CBM type is soft sequential but I don’t see any results on that.
>
> We thank the reviewer for the suggestion. We trained the CBMs on CUB using the soft sequential (SS) and soft independent (SI) versions, and we present the results below. As demonstrated, our regularizers perform as expected and continue to outperform the corresponding baselines.
>
> We will include these results in the final version of the paper. However, we would like to highlight that, based on our experience, most papers report the soft joint version of CBM, with only the original paper presenting the soft sequential versions.
>
> | Method | Concept Acc | Class Acc |
> | -------- | -------- | -------- |
> | CBM SI | $0.956 \pm 0.001$ | $0.639 \pm 0.001$ |
> | **CBM SI + IB_B** | $0.957 \pm 0.001$ | $0.647 \pm 0.003$ |
> | **CBM SI + IB_E** | $0.957 \pm 0.001$ | $0.648 \pm 0.004$ |
> | CBM SS | $0.956 \pm 0.001$ | $0.640 \pm 0.001$ |
> | **CBM SS + IB_B** | $0.957 \pm 0.001$ | $0.670 \pm 0.004$ |
> | **CBM SS + IB_E** | $0.957 \pm 0.001$ | $0.668 \pm 0.002$ |
>
>
> ---
> > What are the leakage results (Table 2) of IB_E?
>
> We report the missing IB_E values on Table 2 below.  We will add them to the final version of the paper.
>
> | Model               | Complete CS - OIS        | Complete CS - NIS        | Selective Drop-out CS - OIS | Selective Drop-out CS -  NIS | Random Drop-out CS - OIS   | Random Drop-out CS - NIS   |
> |---------------------|------------------------|------------------------|---------------------------|---------------------------|--------------------------|--------------------------|
> | CBM (SJ)            | $4.69 \pm 0.43$        | $66.25 \pm 2.31$       | $16.29$                   | $78.39$                   | $12.97 \pm 0.78$         | $74.19 \pm 1.04$         |
> | **CBM (SJ) + IBB**  | $2.16 \pm 0.13$        | $61.67 \pm 1.92$       | $13.09$                   | $73.40$                   | $10.59 \pm 1.48$         | $71.38 \pm 0.89$         |
> | **CBM (SJ) + IBE**  | $2.10 \pm 0.17$        | $62.36 \pm 2.04$       | $13.22$                   | $72.88$                   | $11.00 \pm 1.45$         | $72.03 \pm 1.05$         |
> | CEM                 | $8.74 \pm 0.30$        | $75.41 \pm 3.83$       | $20.85$                   | $80.19$                   | $18.31 \pm 0.09$         | $76.56 \pm 2.00$         |
> | **CEM + IBB**       | $6.11 \pm 0.24$        | $70.02 \pm 2.21$       | $17.22$                   | $76.67$                   | $14.10 \pm 0.42$         | $72.68 \pm 2.38$         |
> | **CEM + IBE**       | $6.14 \pm 0.31$        | $71.28 \pm 1.90$       | $18.00$                   | $75.84$                   | $14.02 \pm 0.37$         | $73.88 \pm 1.97$         |
> | AR-CBM              | $3.90 \pm 0.27$        | $62.30 \pm 1.52$       | $14.16$                   | $63.40$                   | $12.58 \pm 0.86$         | $60.86 \pm 1.32$         |
> | **AR-CBM + IBB**    | $2.83 \pm 0.27$        | $59.87 \pm 1.52$       | $10.97$                   | $59.72$                   | $10.20 \pm 0.55$         | $56.28 \pm 1.33$         |
> | **AR-CBM + IBE**    | $2.70 \pm 0.19$        | $60.04 \pm 0.98$       | $11.05$                   | $60.85$                   | $11.03 \pm 1.02$         | $57.21 \pm 0.85$         |
> | ProbCBM             | $4.30 \pm 0.10$        | $64.22 \pm 1.04$       | $16.01$                   | $76.92$                   | $13.81 \pm 0.21$         | $75.01 \pm 0.86$         |
> | **ProbCBM + IBB**   | $2.53 \pm 0.46$        | $60.35 \pm 2.01$       | $13.11$                   | $72.86$                   | $10.34 \pm 0.55$         | $70.96 \pm 1.91$         |
> | **ProbCBM + IBE**   | $2.61 \pm 0.53$        | $59.86 \pm 1.83$       | $13.07$                   | $73.06$                   | $11.20 \pm 0.68$         | $71.20 \pm 2.07$         |
>
>
> ---
> > Is the number for Table 3 AUC IB_B with 8 corrupt correct? Seems not in line with the trends?
>
> We checked the results again and the reported values are correct. We think that what the reviewer observed is an outlier.  We actually think that the outlier result is the one for 16 concepts instead.
>
> ---
> > Not convinced by the measure of concept-set goodness(4.4), seems like this is closely connected to CBM quality and in the paper it is used to compare different CBMs with the same concept sets, not different concept sets. If the goal is to measure the concept set quality I think the metric should not depend on a particular CBM model.
>
> We acknowledge that the intervention-based metrics introduced in Section 4.4 inherently depend on the specific CBMs used for evaluation, thus **reflecting both concept set quality and model-specific interpretability**. It is true that an ideal 'intrinsic' measure of concept set quality should be independent of modeling choices; our metrics, however, explicitly prioritize measuring the practical interpretability and operational utility of concept sets in realistic intervention scenarios by a given model.
>
> More specifically, **our metrics capture how effectively corrections to concept predictions translate into improved downstream task performance in a given model**, providing critical insights into real-world usage. For tasks involving interpretability and human-in-the-loop interactions, this practical aspect of concept quality (captured by model-specific metrics) is crucial.
>
> Nevertheless, we fully agree that future research should develop complementary metrics aimed explicitly at intrinsic concept set quality, independently of particular CBMs or modeling choices. Such metrics could further enrich our understanding and assessment of concept sets, helping researchers select or refine concept sets independent of the model.
>
> ---
> > Brief introduction in main text would be useful for several things introduced such as the datasets and the Concept Leakage metrics would be helpful.
>
> We thank the reviewer for the suggestion.  However, due to limited space we had to move these definitions into the appendix.  We will review the text to make it clearer that they are in the appendix to help the reader.
>
> ---
> > The introduction categorization into four groups is confusing, what exactly are the four groups? Also seems like these groupings are not mutually exclusive, one method could belong to multiple of these groups, for example post-hoc explanations can be model-agnostic or model-specific.
>
> These groups were our attempt to create a taxonomy to explain the state of the art and to simplify the comparison of our method within the existing ones.  We are aware that this taxonomy is not exhaustive and is polythetic.  We will review our description to ensure that the reader is aware of this limitation.
>
> ---
> > Fig 3 has too many plots in one small fig making it impossible to see all the lines
>
> We agree with the reviewer and have therefore included the individual plots in the appendix (Fig. C.1). However, due to page limitations in the main paper, we had to strike a balance between comprehensively presenting the results and maintaining readability.
>
> ---
> > Some qualititative examples showing interpretable decisions on different datasets would be helpful
>
> We will present qualitative examples from our experiments to illustrate the predicted concepts in the final version of the paper.
>
> ---
> > In eq. 5 I don’t see any term corresponding to the I(X; C) term, is this correct? If yes why is this the case?
>
> Our objective is to transform the mutual information from the information bottleneck (2) into entropies instead (see Eq. A.1).  The idea in this version of the bound is to understand the relation of the mutual information between the data and the concepts through the latent representations and its entropies instead, thus the lack of the term.

---

> > ### Comment · Reviewer_NahS · 2025-08-06
> >
> > Thank you for the response. While this has clarified some of my questions, reading the response and in particular discussion with Reviewer qi7a has significantly reduced my trust in the reported results. In particular it seems clear that at least some of the reported results were using a different backbone than the one mentioned in the paper, and that the baselines were very poorly optimized. If you can improve CBM accuracy by over 10% by training for more epochs then this is clearly not a fully trained model and improving accuracy by 2% over a poorly optimized model is not an interesting finding. I believe the experiments need to be redone with tuned hyperparameters and correct backbones. As a result I am decreasing my score to 3.

---

> > > ### Author Response · Authors · 2025-08-06
> > >
> > > > In particular it seems clear that at least some of the reported results were using a different backbone than the one mentioned in the paper,
> > >
> > > We clarify that **all the baselines and their regularized versions used the same backbone as in their original descriptions**. We apologize for any confusion during the rebuttal, and we will improve the phrasing in the final version of the manuscript. Nevertheless, we emphasize that the results are correct and fair.
> > >
> > > > and that the baselines were very poorly optimized.
> > >
> > > Our objective with the paper is to **demonstrate that our IB-regularizer enhances existing CBM architectures both for prediction and for information leakage reduction**. Since the setup used for each baseline and the regularized versions is consistent, the experiments offer a fair comparison to demonstrate our improvement.
> > >
> > > > If you can improve CBM accuracy by over 10% by training for more epochs then this is clearly not a fully trained model
> > >
> > > Our focus was on understanding the effects of the Information Bottleneck and information leakage, with comparable baselines to regularized models being our primary concern. **Even when models were trained for 1000 epochs, our regularizers showed improvement over baseline models** (as shown in the tables here: https://openreview.net/forum?id=SjAHFGoUb6&noteId=JWwwooe5X3).
> > >
> > > Furthermore, **the definition of a "fully trained model" is unclear**, as there is no guarantee that 1000 epochs is optimal. Other CBMs follow different protocols; for example, CEM trains for 300 epochs while AR-CBM trains for 200.
> > >
> > > Thus, we emphasize that **our comparisons are meaningful and demonstrate the improvements of the IB-regularizers regardless of training length**.
> > >
> > > > and improving accuracy by 2% over a poorly optimized model is not an interesting finding.
> > >
> > > Our contributions lie in introducing a first-principled regularizer using the Information Bottleneck, consistently improving accuracy for concepts and class predictions while reducing leakage. **Focusing solely on the improvement amount over one model class** (the original CBM family) **misrepresents the broader results**, including more recent models like CEM, ProbCBM, and AR-CBM, and the other results for leakage reduction as well as information plane dynamics.
> > >
> > > > I believe the experiments need to be redone with tuned hyperparameters and correct backbones.
> > >
> > > **The baselines used the correct backbones, adhering to each of their original paper' hyperparameters and setups**. Re-doing them will yield the same results as the ones presented in the paper. The discussion about the backbone used for CBM experiments was a misunderstanding, and we apologize for any confusion.
> > >
> > > > As a result I am decreasing my score to 3.
> > >
> > > We hope the reviewer will reconsider their score based on the above clarifications. We believe **our paper presents a significant contribution to the field of concept-based models, and the results are fair and meaningful**. We are open to further clarifications or discussions that may assist in understanding our contributions better.

---

> > > > ### Comment · Reviewer_NahS · 2025-08-07
> > > >
> > > > Thank you for the response, however I will maintain my rating. As I mentioned in the initial comment, improving over a poorly optimized baseline is easy and can be caused by many different reasons beyond the method being better. For good science the hyperparameters of all baselines should be optimized and the authors should test whether for example increasing the epochs significantly improves performance. If training for 2000 epochs gives similar results to 1000 epochs that would be good evidence towards models being "fully trained".
> > > >
> > > > > We clarify that all the baselines and their regularized versions used the same backbone as in their original descriptions
> > > >
> > > > I'm still a little confused on this point. Can you point out exactly which backbone was used for each of the results you report in Table 1?

---

> > > > > ### Author Response · Authors · 2025-08-07
> > > > >
> > > > > > For good science the hyperparameters of all baselines should be optimized and the authors should test whether for example increasing the epochs significantly improves performance. If training for 2000 epochs gives similar results to 1000 epochs that would be good evidence towards models being "fully trained".
> > > > >
> > > > > Indeed, all our choices have been made to ensure good science. Specifically, we reproduced results from previous work using their exact setups, including the number of training epochs. The only exception was the CBM model, where we followed the baseline from the original paper but adopted the number of training epochs from the ProbCBM paper. We did not consider this a problem because the IB-regularized version of this model was trained in precisely the same way, using the exact same baseline. Nevertheless, we demonstrated that even when trained for 1000 epochs, the relative improvement remains the same (see results below).
> > > > >
> > > > > | Method | Epochs | Concept | Class |
> > > > > | - | - | - | - |
> > > > > | CBM | 100 | 0.956 | 0.708 |
> > > > > | CBM + IB-B | 100 | 0.958 | 0.725 |
> > > > > | CBM + IB-E | 100 | 0.959 | 0.729 |
> > > > > | **CBM** | 1000 | 0.957 | 0.795 |
> > > > > | **CBM + IB-B** | 1000 | 0.957 | 0.820 |
> > > > > | **CBM + IB-E** | 1000 | 0.957 | 0.822 |
> > > > >
> > > > > Please note that in all other cases, we adhered to the same baselines and hyperparameters to ensure reproducibility of the published results, and applied them likewise in their respective IB versions.
> > > > >
> > > > > Secondly, we **assert that improvements in performance should not be judged solely on the basis of improvements in class label prediction accuracy**. Instead, it should be evaluated as a combination of: (a) stable concept label prediction, (b) improved class label prediction, (c) enhanced interpretability, and (d) better intervention capacity. In all cases, the IB-regularized versions have demonstrated superior performance across all of these four factors.
> > > > >
> > > > > > I'm still a little confused on this point. Can you point out exactly which backbone was used for each of the results you report in Table 1?
> > > > >
> > > > > The specific details for each model are:
> > > > > - For the CBM family, the original paper used Inception, and we did the same, although our code is sourced from the ProbCBM paper.
> > > > > - ProbCBM employed ResNet18, and we utilized their configuration (and backbone) and code for our experiments.
> > > > > - CEM used ResNet34 in their paper, which we also adopted, using code from the CEM repository.
> > > > > - AR-CBM used Inception, which we followed, using code from the AR-CBM repository.

---

> > > > > > ### Comment · Reviewer_NahS · 2025-08-09
> > > > > >
> > > > > > Thank you for the response.
> > > > > >
> > > > > > This has clarified some of my concerns and given me more faith in the reported results. However I still have considerable uncertainty about the reliability of the experimental results, and whether the experimental choices are the best for comparing different methods (i.e. comparing different backbones in the same table without mentioning) and some poor hyperparameter choices in original experiments, and I would need to be able to review the full rewritten manuscript to reach confidence in the results, which is unfortunately not possible with this year's rebuttal format.
> > > > > >
> > > > > > I believe this paper makes a useful contribution but the evidence I have seen for it does not currently meet the standard for conference acceptance in my opinion. I recommend the authors re-evaluate the experiments to make sure hyperparameters are tuned well and add some comparisons where the backbones are the same across different methods.
> > > > > >
> > > > > > As such I cannot recommend acceptance and will maintain my rating of borderline reject.

---

> > > > > > > ### Author Response · Authors · 2025-08-09
> > > > > > >
> > > > > > > > whether the experimental choices are the best for comparing different methods (i.e., comparing different backbones in the same table without mentioning) and some poor hyperparameter choices in original experiments
> > > > > > >
> > > > > > > We reiterate that we conducted a fair comparison by using the same setup for each baseline and then applying our IB-regularizer. **This approach is the most effective for ensuring a fair comparison and clearly demonstrating the improvement brought by our proposal, as all other variables remain constant.** Furthermore, we ensured that each baseline is run with the authors' recommended backbone and hyperparameters to reproduce its intended performance.
> > > > > > >
> > > > > > > We will add details about the different setups and our findings during the rebuttal to strengthen the discussion. However, **we disagree that failing to report these setups initially is equivalent to conducting poor experiments**.
> > > > > > >
> > > > > > > > and I would need to be able to review the full rewritten manuscript to reach confidence in the results, which is unfortunately not possible with this year's rebuttal format.
> > > > > > >
> > > > > > > It is indeed unfortunate. However, we assure you that the discussion will be a summarized version of the facts presented in this rebuttal. Moreover, as previously mentioned, the experimental results will remain unchanged, as they are correct and align with the setup of each baseline.
> > > > > > >
> > > > > > > > I recommend the authors re-evaluate the experiments to ensure hyperparameters are tuned well and add some comparisons where the backbones are consistent across different methods.
> > > > > > >
> > > > > > > We appreciate the reviewer's suggestion. While such an experiment could offer insights into which backbone is most relevant for CBMs, **it aligns with a different line of research than our proposal**.
> > > > > > >
> > > > > > > Our **primary focus is to investigate the impact of IB-regularizers** on CBMs concerning accuracy, information leakage, and representations. *Using a single backbone across all methods would create straw-man baselines*, as some methods have been tuned and validated on specific architectures and feature dimensions. Furthermore, our claim is intra-method; for every method X, we compare X to X-IB using the same backbone, data, and training recipe. **This approach isolates the effect of the IB regularizer**, which is the scientific claim of the paper.
> > > > > > >
> > > > > > > > I believe this paper makes a useful contribution but the evidence I have seen for it does not currently meet the standard for conference acceptance in my opinion.
> > > > > > >
> > > > > > > We appreciate the reviewer's positive comments regarding our contribution. However, we disagree with the assertion of insufficient evidence, **as the experimental results are clearly presented and demonstrate improvement across multiple metrics**. We understand that the misunderstanding during our discussion with reviewer qi7a may have led to confusion for others as well, and we will address this in the final discussion. We emphasize that even if we follow the reviewer's advice to re-run the experiments, the results will remain unchanged, as the hyperparameters are correct and aligned with the original setups proposed by the authors of each baseline.
> > > > > > >
> > > > > > >
> > > > > > > Moreover, we emphasize that our proposal is the **first to introduce and integrate the Information Bottleneck framework into CBMs**. Therefore, *the combination of theoretical contributions and empirical evidence presents compelling results for the conference*. We remind the reviewer that NeurIPS encourages the presentation of novel ideas over mere percentage improvements in results. Consequently, we believe that our proposal is well-suited for presentation at NeurIPS.

---

> ### Comment · Area_Chair_6Jph · 2025-08-05
> **Ping**
>
> Dear Reviewer,
>
> The deadline for the author-reviewer discussion is approaching (Aug 8, 11.59pm AoE).
> Please read carefully the authors' rebuttal and engage in meaningful discussion.
>
> Thank you,
> Your AC

---

> ### Comment · Reviewer_qi7a · 2025-08-07
>
> >The specific details for each model are:
>
> >For the CBM family, the original paper used Inception, and we did the same, although our code is sourced from the ProbCBM paper.
> >ProbCBM employed ResNet18, and we utilized their configuration (and backbone) and code for our experiments.
> >CEM used ResNet34 in their paper, which we also adopted, using code from the CEM repository.
> AR-CBM used Inception, which we followed, using code from the AR-CBM repository.
>
> **The major concern for me is that the ProbCBM paper clearly states that all experiments were conducted using ResNet18 (including CBM, CEM, and ProbCBM), yet the results in your paper are exactly identical to theirs (including CBM, CEM, and ProbCBM).** There are three possible explanations as I see it:
>
> 1. You may have directly copied the results from the ProbCBM paper.
>
> 2. The ProbCBM paper might have actually used three different backbones (e.g., ResNet18, ResNet34, Inception) when conducting experiments, but mistakenly claimed in the text that all experiments were conducted using ResNet18.
>
> 3. A very remarkable coincidence happened. The CBM results you obtained using Inception are exactly identical to the CBM results reported in the ProbCBM paper with ResNet18 backbone, and the CEM results you obtained using ResNet34 are exactly identical to the CEM results reported in the ProbCBM paper (using ResNet18).
>
> If the authors are not intentionally misleading, then only the second and third possibilities remain. I would like to ask the authors what they think is the actual situation. Of course, I also welcome the authors to propose any other possible explanations.
>
> Finally, I want to make it clear that I am not intending to give a negative evaluation. I truly want to understand what happened because the current results are quite confusing to me. If it turns out that your experiments are indeed correct and properly conducted, I am willing to revise my score accordingly.

---

> > ### Author Response · Authors · 2025-08-08
> >
> > > The major concern for me is that the ProbCBM paper clearly states that all experiments were conducted using ResNet18 (including CBM, CEM, and ProbCBM), yet the results in your paper are exactly identical to theirs (including CBM, CEM, and ProbCBM).
> >
> > We understand the reviewer's concern and assure you that all our experiments, particularly the setups, were conducted in good faith to deliver fair and comparable results.
> >
> > Regarding ProbCBM, our results follow those of ResNet18, aligning with the original ProbCBM paper.
> >
> > In the case of CEM and CBM, despite the claims in ProbCBM, we believe there are inconsistencies between their reported results and the claims regarding the backbones, as explained below.
> >
> > For CEM, the results are remarkably similar between ResNet18 and ResNet34, as noted by the CEM authors in Appendix A.4 (Figure A.7) of the CEM paper. This similarity might explain the scores reported in the ProbCBM paper. Additionally, there is no evidence that the authors of the ProbCBM paper implemented and ran CEMs; we rely on their assertion regarding ResNet18, but there is no code-based justification. We executed CEM with 300 epochs, as described in the original paper, using different backbones and found accuracies of 0.757, 0.762, and 0.770 for ResNet18, ResNet34, and Inception, respectively. These results support the original CEM paper's claim that there is no significant difference with respect to the backbone. Moreover, the results for the ResNet models fall within the standard deviation of the values reported in the ProbCBM paper. However, establishing significant results would require more runs and a proper hypothesis test, which are beyond the scope of this rebuttal.
> >
> > Regarding CBM results, the performance between Inception and ResNet18 backbones is similar, albeit slightly lower with ResNet18. For example, in the CUB dataset trained for 100 epochs, we observed a class accuracy of 0.708 with the Inception backbone compared to 0.683 with ResNet18. Considering our experiments with up to 1000 epochs, we predict these results will scale similarly. Notably, the concept accuracies are comparable at 0.95, suggesting that the results reported in the ProbCBM paper align more closely with the Inception backbone than with ResNet18.
> >
> > We emphasize that in our paper the baselines and the IB-regularized versions utilized the same setups (backbones and hyperparameters). Additionally, we validated the execution of the base model and found that our results were within one standard deviation of the reported baselines in the original papers, confirming the accuracy of our baselines. Despite these sanity check values, we reported the original values, as is standard practice.
> >
> > > If it turns out that your experiments are indeed correct and properly conducted, I am willing to revise my score accordingly.
> >
> > We reiterate that the experiments were conducted correctly and properly. We appreciate that the reviewer is willing to revise their score.

---

### Note · Authors · 2025-08-14

Our research advances Concept Bottleneck Models (CBM) by proposing an Information Bottleneck (IB) regularization, addressing key challenges with **solid, systematic, and extensive empirical and theoretical evidence**.  We were the **first ones** to make this link and formalize it.  Our results combine quantitative metrics, intervention experiments, and theoretical analyses enables us to present comprehensive evidence of leakage reduction, offering valuable insights into model interpretability.

**Main issues:**
1. *Resolving Prior Inconsistencies*: The discrepancy between the reported results and prior CBM work was due to difference in the trained epochs.  We showed the results and demonstrated that our experiments reproduce the baselines, but also improves them regardless of the length of training.  We highlight that our evaluation setup follows the baselines and presents comparable and fair experiments.  We discuss the differences that may be present in the reported results from our discussion with *qi7a*.  At the end of the discussion, *qi7a* acknowledged our experimental setup and increased their score.

2. *Beyond Marginal Gains*: Our comprehensive evaluations reveal improvements not only in accuracy but also in information leakage reduction and information plane dynamics analysis. These broader benefits extend beyond individual model enhancements.

3. *Focus on Generalization*: We emphasize that "fully trained models" is not clearly defined, and that recent models report results for short trainings (sub 300 epochs).  Longer training prioritizes benchmark over-fitting over understanding the impact of the proposal w.r.t. the baselines.

4. *Comparison and Leakage Analysis*: We clarified evaluations against existing leakage methods, proving enhanced reduction across various CBM versions, answering newly raised concerns.

5. *Motivation and Parameter Choice*: We refined the motivation for H(C) and provided empirical evidence for simplification. Our selection of the hyperparameter β is backed by robust ablation studies.

---

### Decision · Program_Chairs · 2025-09-17

**Decision:**

Reject

**Comment:**

TBD